# Acetylation is required for full activation of the NLRP3 inflammasome

Yening Zhang[1,2], Ling Luo[1], Xueming Xu[1], Jianfeng Wu[3], Fupeng Wang[1], Yanyan Lu[4], Ningjie Zhang[5], Yingying Ding[6], Ben Lu [1,2] ✉ & Kai Zhao [1,2] ✉

Full activation of the NLRP3 inflammasome needs two sequential signals: a priming signal, followed by a second, assembly signal. Several studies have shown that the two signals trigger post-translational modification (PTM) of NLRP3, affecting activity of the inflammasome, however, the PTMs induced by the second signal are less well characterized. Here, we show that the assembly signal involves acetylation of NLRP3 at lysine 24, which is important for the oligomerization and the actual assembly of NLRP3 without affecting its recruitment to dispersed trans-Golgi network (dTGN). Accordingly, NLRP3 inflammasome activation is impaired in NLRP3-K24R knock-in mice. We identify KAT5 as an acetyltransferase able to acetylate NLRP3. KAT5 deficiency in myeloid cells and pharmacological inhibition of KAT5 enzymatic activity reduce activation of the NLRP3 inflammasome, both in vitro and in vivo. Thus, our study reveals a key mechanism for the oligomerization and full activation of NLRP3 and lays down the proof of principle for therapeutic targeting of the KAT5-NLRP3 axis.

Nucleotide-binding domain, leucine-rich repeat, and pyrin domain–containing protein 3 (NLRP3), which belongs to NLR family, can be activated by a broad range of microbial components, endogenous danger signals and environmental irritants. Upon activation, NLRP3 forms an inflammasome complex with the adapter apoptosis-associated speck like protein containing a CARD (ASC), leading to caspase-1 activation, proinflammatory cytokines release, and cell death. Aberrant NLRP3 inflammasome activation importantly contributes to the pathogenesis of multiple diseases including inflammatory, metabolic, degenerative and aging-related disorders[1–3].

It is well-established that the NLRP3 inflammasome activation requires two sequential steps, namely the priming step and the assembly step. Pathogen-associated molecule patterns (PAMPs), such as LPS, provide the first signal that promotes the expression of NLRP3 and IL-1β. Diverse stimuli, such as ATP and nigericin, a pore-forming toxin derived from *Streptomyces hygroscopicus*, provide the second signal that triggers the assembly of NLRP3 inflammasome[3–5]. Although

several models have been proposed to explain the common pathways of the NLRP3 inflammasome activation in response to different stimuli. These include potassium efflux, mitochondrial damage, ROS production, lysosomal disruption and Trans-Golgi disassembly. However, it still remains unclear that how these diverse stimuli result in the assembly of NLRP3 inflammasome[3–5]. Further efforts are still needed to elucidate the precise mechanisms by which NLRP3 is activated by numerous distinct stimuli.

Recent studies have provided insights into how the NLRP1B (Nucleotide-binding Domain, leucine-rich repeat and pyrin domain-containing 1B) inflammasome is activated[6,7]. In this context, ubiquitination of the N terminal of NLRP1B promotes the degradation of N terminus, and subsequently liberates the NLRP1B C terminus to form the inflammasome complexes to activate caspase-1. This finding suggests that post-translational modifications (PTMs) may play important role in the inflammasome activation. Although there are several PTMs on NLRP3 were reported, such as ubiquitylation, phosphorylation,

[1]Department of Hematology and Critical Care Medicine, the Third Xiangya Hospital, Central South University, Changsha, Hunan Province, 410000, P. R. China. [2]Key Laboratory of Sepsis Translational Medicine of Hunan, Central South University, Changsha, Hunan Province, 410000, P. R. China. [3]State Key Laboratory of Cellular Stress Biology Innovation Center for Cell Signaling Network, School of Life Sciences, Xiamen University, Xiamen, Fujian Province, 361005, P. R. China. [4]Department of Hematology, The Second Xiangya Hospital, Central South University, Changsha, Hunan Province, 410000, P. R. China. [5]Department of Blood Transfusion, The Second Xiangya Hospital, Central South University, Changsha, Hunan Province, 410000, P. R. China. [6]Department of Pathogen Biology, NavaMedical University, Shanghai 200082, P. R. China. ✉e-mail: xybenlu@csu.edu.cn; kaizhao@csu.edu.cn

sumoylation and palmitoylation[4,5,8], these studies mostly focused on the PTMs induced by the priming signal, the second signal induced PTMs on NLRP3 and associated activities changed remains less defined.

Here, we report that NLRP3 is acetylated upon activation signal. Further experiments demonstrate that lysine acetyltransferase5 (KAT5) mediates acetylation of NLRP3 on lysine 24, which promotes NLRP3 oligomerization without affecting its recruitment to dispersed *trans*-Golgi network(dTGN). Accordingly, KAT5 deficiency diminishes the NLRP3 inflammasome activation in vitro and in vivo. Pharmacological inhibition of KAT5 by NU 9056 blocks the NLRP3-dependent inflammatory responses, providing potential for the treatment of NLRP3 associated diseases. Thus, this study uncovers that acetylation is critical for the activation of NLRP3 inflammasome in response to diverse stimuli.

## Results

### NLRP3 is acetylated during inflammasome activation

Since acetylation plays a pivotal role in various biological processes[9,10], we asked whether it participates in the NLRP3 inflammasome activation. We detected the overall acetylation levels in mouse primary macrophages stimulated with LPS alone, LPS + ATP, or LPS+nigericin

by antibodies against acetylated lysines, we observed that the allover acetylation levels of proteins were obviously increased upon LPS + ATP or LPS+nigericin stimulation (Supplementary Fig. 1a), suggesting that acetylation may have a regulatory role in the NLRP3 inflammasome activation. Intriguingly, we noticed that acetylation levels changed significantly between 55 Kd and 130 Kd, in which NLRP3 is within the molecular weight. Thus, we decided to determine whether NLRP3 is acetylated. By stimulated primary macrophages with different NLRP3 inflammasome stimuli, including ATP (K⁺-efflux dependent), MSU (lysosome disruption dependent) and Imiquimod (K⁺-efflux independent)[3], we found that NLRP3 was acetylated in response to the activation signal (Fig. 1a), but not the priming signal (Fig. 1a). Transfection of macrophages with poly(dA:dT), an agonist of absent in melanoma 2 (AIM2) inflammasome, or flagellin, a NLR family, CARD domain containing 4 (NLRC4) inflammasome activator, did not induce NLRP3 acetylation (Fig. 1b). These data demonstrated that NLRP3 undergoes acetylation during inflammasome activation, which is in line with previous study[11].

### Identification of the acetylation sites in NLRP3

Next, we utilized an acetyl proteomic approach to identify the acetylation sites within NLRP3. We first reconstituted the NLRP3

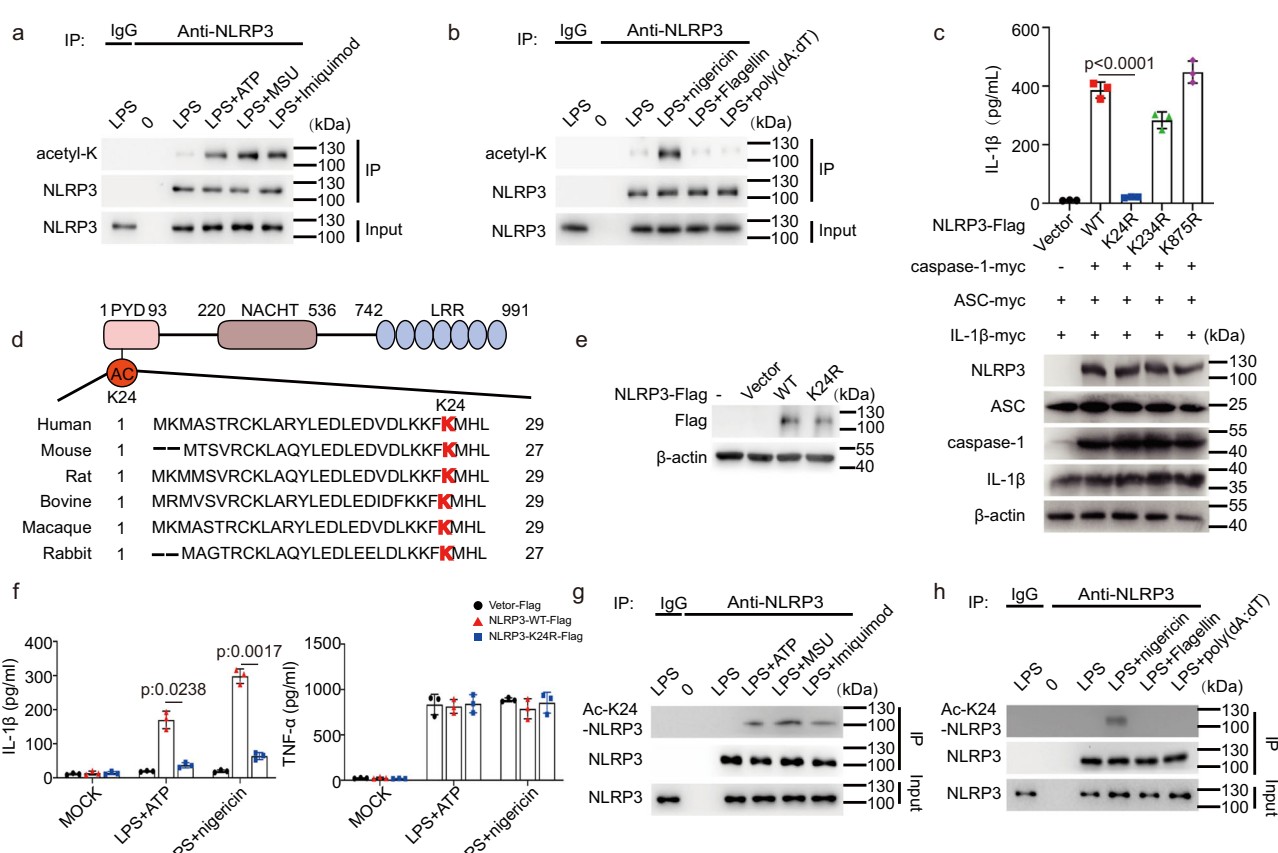

**Fig. 1 | NLRP3 Acetylation at Lys24 is indispensable for NLRP3 inflammasome activation. a**, **b** Immunoblot analysis of acetylation level of NLRP3 from peritoneal macrophage treated with LPS (100 ng/mL, 3 h) along or with ATP (5 mM, 1 h), MSU (200 μg/mL, 6 h), Imiquimod (40 ug/mL, 1 h), nigericin (10 μM, 1 h), Flagellin transfection (2 μg/mL, 1 h) or poly (dA:dT) transfection (1 μg/mL, 16 h). **c** ELISA of IL-1β in supernatants and Immunoblot analysis of whole cell lysis from HEK293T cells reconstituted with NLRP3 inflammasome and stimulated with nigericin (10 μM, 1 h). *n* = 3 biologically independent experiments. **d** Scheme of NLRP3 domain structure and the alignment of NLRP3 orthologs. The acetylated lysine residue is underlined in red. **e**, **f** NLRP3⁻/⁻ iBMDMs were reconstituted with WT or K24R NLRP3-Flag, then treated with LPS (1 μg/mL, 6 h) along or with ATP (10 mM, 1 h), nigericin (20 μM, 1 h).

**e** Cell Lysates were subjected to western blot analysis. **f** ELISA analysis of IL-1β and TNF-α in supernatants. *n* = 3 biologically independent experiments.

**g**, **h** Immunoblot analysis of K24 acetylation level of NLRP3 from peritoneal macrophage treated with LPS (100 ng/mL, 3 h) along or with ATP (5 mM, 1 h), MSU (200 μg/mL, 6 h), Imiquimod (40 ug/mL, 1 h) or nigericin (10 μM, 1 h), Flagellin transfection (2 μg/mL, 1 h) or poly (dA:dT) transfection (1 μg/mL, 16 h). Results are represented as mean ± SD and typical photographs are representative of three biological independent experiments with similar results. Statistical analyses were carried out via one-way ANOVA with Dunnett's test for (**c**) and two-way ANOVA with the Bonferroni test for (**f**). Source data are provided as a Source Data file.

inflammasome in HEK293T cells by transfecting NLRP3, ASC, caspase-1 and Pro-IL-1β plasmids, followed by nigericin stimulation. Since maturation of IL-1β is regarded as an indicator of NLRP3 inflammasome activity (Supplementary Fig. 1b), this reconstitution system provided a relevant model to study NLRP3 inflammasome[12,13]. Flag-tagged NLRP3 was immunoprecipitated and analyzed by liquid chromatography-mass spectrometry. Three acetylation sites (Lys24, Lys234 and Lys875) were identified (Supplementary Fig. 1c). Among them, Lys234 and Lys875 were the common sites in the two groups (non-treated and nigericin treated groups), while the Lys24 was exclusively identified after nigericin stimulation. To determine the roles of these acetylation sites in NLRP3 inflammasome activation, we replaced lysine at position 24, 234, or 875 with arginine (K24R, K234R or K875R) to mimic nonacetylated state of these lysine residues. The biological function of each NLRP3 variant was examined in HEK293T cells reconstituted with the NLRP3 inflammasome. The K24R NLRP3 mutants were unable to promote pro-IL-1β cleavage, while others have the similar promoting effect as wild-type NLRP3 (Fig. 1c). The Lys24 residue, the unique acetylation site identified in nigericin treated group, is located in the pyrin domain (PYD) and conserved across many species (Fig. 1d). To further address the functional importance of Lys24 in the activation of NLRP3 inflammasome, we re-expressed of NLRP3-K24R or NLRP3 wild type (WT) in NLRP3[-/-] Immortalized bone marrow-derived macrophages (iBMDMs). Though WT NLRP3 and NLRP3-K24R were expressed at the comparable level (Fig. 1e), after stimulation with ATP or nigericin, the release of IL-1β, but not TNF-α, was markedly reduced in the NLRP3-K24R-expressing cells, as compared to WT NLRP3-expressing cells (Fig. 1f). To further confirm that NLRP3 undergoes lys24 acetylation, we generated a specific polyclonal antibody against NLRP3 acetylated at Lys24. Dot blot analysis showed that this antibody specifically recognized NLRP3 peptides acetylated at Lys24, but not un-acetylated peptides or NLRP3 peptides acetylated at Lys21 or Lys22 (Supplementary Fig. 1d), both of which were recently identified acetylation sites in NLRP3[11]. Overexpression experiments further confirmed the specificity of this antibody (Supplementary Fig. 1e). Then, this produced antibody was used to tested acetylation of NLRP3 at Lys24. As shown by western-blot of primary macrophages, acetylation of NLRP3 at Lys24 was enhanced when treated with ATP and Imiquimod, but not poly(dA:dT) or flagellin transfection. (Fig. 1g, h). Collectively, these results suggested lys24 acetylation is critical for the NLRP3 inflammasome activation.

## Lys24 acetylation of NLRP3 promotes the inflammasome activation

To evaluate the function of NLRP3 Lys24 acetylation in physiological setting, we generated knock-in (KI) mice harboring the Nlrp3 K24R allele (Nlrp3[K24R/ K24R]), in which the Lysine-24 was replaced by arginine (Supplementary Fig. 2a, b), we observed that the mutation did not affect the expression of NLRP3 (Supplementary Fig. 2c). Consistent with the results from NLRP3-reconstituted iBMDMs (Fig. 1f), Nlrp3[K24R/ K24R] bone marrow-derived macrophages (BMDMs) exhibited impaired NLRP3 inflammasome activation, but not AIM2 or NLRC4 inflammasome activation (Fig. 2a–c).

We next assessed the function of NLRP3 Lys24 acetylation in vivo, we adopted an LPS-induced endotoxin shock model. After LPS challenge, the levels of IL-1β, but not TNF-α in serum were significantly reduced in Nlrp3[K24R/K24R] mice, as compared to that of in their WT littermates (Fig. 2d, e). The alleviated lung injury was also observed in Nlrp3[K24R/K24R] mice (Histological examination of lungs and evaluation of lung injury scores[14]) (Fig. 2f, g). Western Blot analysis revealed lower levels of cleaved caspase-1 and IL-1β in lungs of Nlrp3[K24R/ K24R] as compared to that of in WT littermates (Fig. 2h). Together, these results indicate that Lys24 acetylation of NLRP3 promotes the inflammasome activation in vitro and in vivo.

## Lys24 acetylation of NLRP3 regulates NLRP3 oligomerization

Then, we investigated how Lys24 acetylation of NLRP3 regulates the inflammasome activation. Since Lys24 is located in the PYD domain of NLRP3, which contributes to the assembly of the inflammasome by connecting NLRP3 and ASC[2,3], we tested whether Lys24 acetylation of NLRP3 affects the NLRP3 and ASC interaction. We first transfected cells with plasmids encoding those NLRP3 mutants and observed the interrupted interaction between NLRP3-K24R and ASC compared with NLRP3-K234R, NLRP3-K875R, and NLRP3 WT (Fig. 3a). Similar results were observed in NLRP3-NEK7 interaction (Fig. 3b), which is an essential factor for NLRP3 inflammasome assembly[15–17]. Oligomerization of NLRP3 is an earlier event for the inflammasome assembly[3,18], we also observed impaired interaction between Flag-NLRP3 and Myc-NLRP3 when K24R mutant was introduced (Fig. 3c). Moreover, semi-denaturing detergent agarose gel electrophoresis (SDD-AGE), a method for detecting large protein oligomers in studying prions[19,20], blue native polyacrylamide gel electrophoresis (BN-PAGE), a method for detecting large native protein complexes in the mass range of 10 KDa to 10 MDa[21], and live cell imaging were adopted to detect the NLRP3 oligomerization. In agreement with previous studies[20,22,23], NLRP3 formed large oligomers upon nigericin stimulation, and the oligomerization was markedly inhibited by the K24R mutation of NLRP3 (Fig. 3d). Further, live cell imaging revealed that K24R NLRP3 exhibited smaller and thinner puncta when stimulated with nigericin as compared to WT NLRP3 (Fig. 3e, f and supplementary Movie 1,2 and 3). Moreover, reduced formation of NLRP3 oligomers were observed in Nlrp3[K24R/ K24R] macrophages compared with WT cells by SDD-AGE (Fig. 3g). To better reveal the formation of NLRP3 oligomers, BN-PAGE, an extensively applied approach to detect the oligomerization of NLRP3, were adopted[16,24]. The digitonin-solubilized cell lysates were separated in the first dimension by BN-PAGE followed by a second dimension of SDS-PAGE. Consistent with previous studies[16,24], high-molecule-weight complexes containing NLRP3 were observed in WT macrophages upon LPS plus ATP or nigericin stimulation. The formation of these complexes were markedly impaired in Nlrp3[K24R/K24R] macrophages (Fig. 3h, i). Thus, NLRP3 K24 acetylation promotes the NLRP3 oligomerization during inflammasome activation.

Recent work indicated that the dispersed TGN (dTGN) is a platform for NLRP3 aggregation into multiple puncta[25]. To investigate whether K24 acetylation in NLRP3 influence this process, we monitored the co-localization between NLRP3 and dTGN in cells. We overexpressed GFP, GFP-tagged WT NLRP3 or GFP-tagged K24R NLRP3 in Cos7 cells, and TGN38 and GOLGA4, markers of TGN were stained. Upon nigericin stimulation, the number of TGN38 or GOLGA4 fragments did not change among three groups (Supplementary Fig. 3a–d). Immunofluorescence analysis revealed that K24R mutation has no effects on co-localization between NLRP3 puncta and dTGN(Supplementary Fig. 3a, b, e, f) suggesting that K24 acetylation in NLRP3 neither affect the formation of dTGN nor the recruitment of NLRP3 to dTGN.

## KAT5 mediates the acetylation of NLRP3

Next, we investigated which acetyltransferase is responsible for acetylation of NLRP3. For this purpose, we tested all the commercially available acetyltransferase inhibitors. LPS-primed macrophages were incubated with these inhibitors, and then stimulated with nigericin. Notably, NU 9056 exhibited a robust inhibition of NLRP3 inflammasome at 1 μM, while the others had no such obvious suppressive effect (Supplementary Fig. 4a).Since NU 9056 is an inhibitor of Lysine acetyltransferase 5 (KAT5)[26,27](Supplementary Fig. 4a), we explored the role of KAT5 in NLRP3 acetylation. We first confirmed the direct interaction between KAT5 and NLRP3 by immunoprecipitation, confocal microscopy and GST pull-down assay (Fig. 4a, b, Supplementary Fig. 4b, c). Moreover, co-immunoprecipitation experiments indicated that

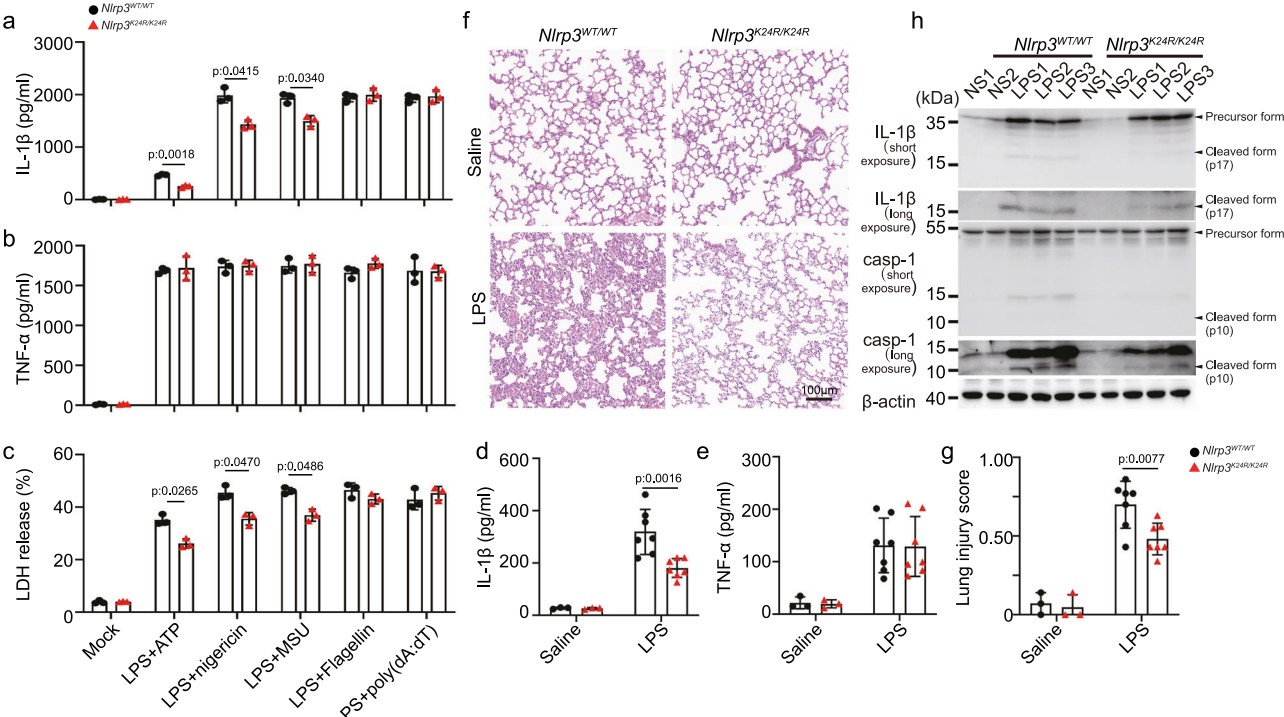

**Fig. 2 | Lys24 acetylation of NLRP3 promotes inflammasome activation.**
**a**–**c** BMDMs from *Nlrp3^WT/WT* (*n* = 3) or *Nlrp3^K24R/K24R* (*n* = 3) mice were treated with LPS (100 ng/mL, 3 h) with ATP (5 mM, 1 h), nigericin (10 μM, 1 h), MSU (200 μg/mL, 6 h), Flagellin transfection (2 μg/mL, 1 h) or poly (dA:dT) transfection (1 μg/mL, 16 h). **a**, **b** ELISA analysis of IL-1β and TNF-α in supernatants. **c** Release of LDH in supernatants. **d**–**h** *Nlrp3^WT/WT* or *Nlrp3^K24R/K24R* male mice of 6–8 weeks were injected with Saline (*n* = 3 biologically independent mice) or LPS (20 mg/kg, i.p.) (*n* = 7 biologically independent mice) for 12 h. **d**, **e** ELISA analysis of IL-1β and TNF-α in serum. **f** representative H&E images lung sections. Scale bar = 100 μm. **g** lung injury score. **h** Immunoblot analysis of IL-1β and caspase-1 from lung of *Nlrp3^WT/WT* or *Nlrp3^K24R/K24R* mice. casp-1(caspase-1). Results are represented as mean ± SD. Statistical analyses were carried out via two-way ANOVA with the Bonferroni test for (**a**–**e**, **g**). Source data are provided as a Source Data file.

endogenous KAT5-NLRP3 interaction was markedly enhanced upon nigericin treatment (Fig. 4a). In contrast, KAT5 did not interact with ASC or caspase-1 (Supplementary Fig. 4d, e). Previous studies[28,29] have shown that KAT5 S86A mutation could substantially attenuate the acetyltransferase activity of KAT5. We obtained a knock-in mouse strain in which the wild-type *Kat5* allele is replaced by the *Kat5-S86A* mutant (serine-86 altered to alanine). We observed decreased acetylation of NLRP3 in primary macrophages upon treated with nigericin from *Kat5^SA/SA* mice compared to its littermates (Fig. 4c). Moreover, knockdown of KAT5 in primary macrophages obtained the similar results (Fig. 4d). Acetylation assay further demonstrated KAT5 could directly acetylate NLRP3 in vitro (Fig. 4e). Taken together, these data implied that KAT5 interacts with NLRP3 and directly mediates the acetylation of NLRP3.

### KAT5 promotes the NLRP3 inflammasome activation through acetylation and oligomerization of NLRP3
To further explore the role of KAT5 in NLRP3 inflammasome activation, we silenced KAT5 in primary macrophages and iBMDMs by siRNA or short hairpin RNAs (shRNAs), respectively. Knockdown of KAT5 remarkably suppressed IL-1β but not TNF-α secretion, caspase-1 maturation and cell death triggered by the NLRP3 inflammasome agonists, but not the AIM2 or NLRC4 inflammasome agonists (Supplementary Fig. 5a–d). To further confirm the phenomenon, we generated *Kat5^fl/fl lyz2-Cre* mice with KAT5 deletion in myeloid cells[30]. Similarly, BMDMs from KAT5-deficient (*Kat5^fl/fl lyz2-Cre*) mice exhibited impaired NLRP3 inflammasome activation, but not AIM2 or NLRC4 inflammasome activation as compared to that of KAT5-sufficient (*Kat5^fl/fl*) mice (Fig. 5a–c). KAT5 deficiency had no effect on the mRNA or protein expression of NLRP3, ASC, caspase-1 or pro-1β (Fig. 5d,

Supplementary Fig. 6). Consistent with the data acquired from the NLRP3-K24R overexpression, acetylation (including K24 acetylation) was sharply declined in the absence of KAT5 (Fig. 5e). SDD-AGE and BN-PAGE analysis revealed that KAT5 deficiency reduced the formation of NLRP3 oligomers (Fig. 5f–h). Besides, macrophages from *Kat5^SA/SA* mice phenocopied *Kat5^fl/fl lyz2-Cre* mice (Supplementary Fig. 7a, b). Thus, these data demonstrated that KAT5 promotes NLRP3 inflammasome activation via controlling acetylation and oligomerization of NLRP3.

### KAT5 promotes NLRP3 inflammasome activation in vivo
To gain insights into the roles of KAT5 in the NLRP3 inflammasome activation in vivo, LPS-induced endotoxin shock model was adopted, upon LPS challenge, the serum levels of IL-1β but not TNF-α were significantly reduced in *Kat5^fl/fl lyz2-Cre* mice, as compared to that of in *Kat5^fl/fl* mice (Fig. 6a, b). The lung tissue also exhibited less damage and infiltration of immune cells (Fig. 6c, d). Western Blot analysis showed lower levels of cleaved caspase-1 and IL-1β in lungs of *Kat5^fl/fl lyz2-Cre* mice than *Kat5^fl/fl* mice (Fig. 6e). Further, in the monosodium urate (MSU) induced peritonitis model, the IL-1β release and neutrophil infiltration in the peritoneal lavage were significantly decreased in *Kat5^fl/fl lyz2-Cre* mice, as compared to that of in *Kat5^fl/fl* mice (Fig. 6f, g, Supplementary Fig. 8). Thus, these data demonstrated that KAT5 is critical for the full activation of the NLRP3 inflammasome in vivo.

### KAT5 inhibitor NU 9056 suppresses the NLRP3 inflammasome activation
Finally, we evaluated the potential of targeting KAT5-mediated acetylation of NLRP3 in alleviating the related inflammatory responses. We observed NU 9056, a KAT5 inhibitor, inhibited the NLRP3 inflammasome activation, but not the NLRC4 or AIM2 inflammasome in

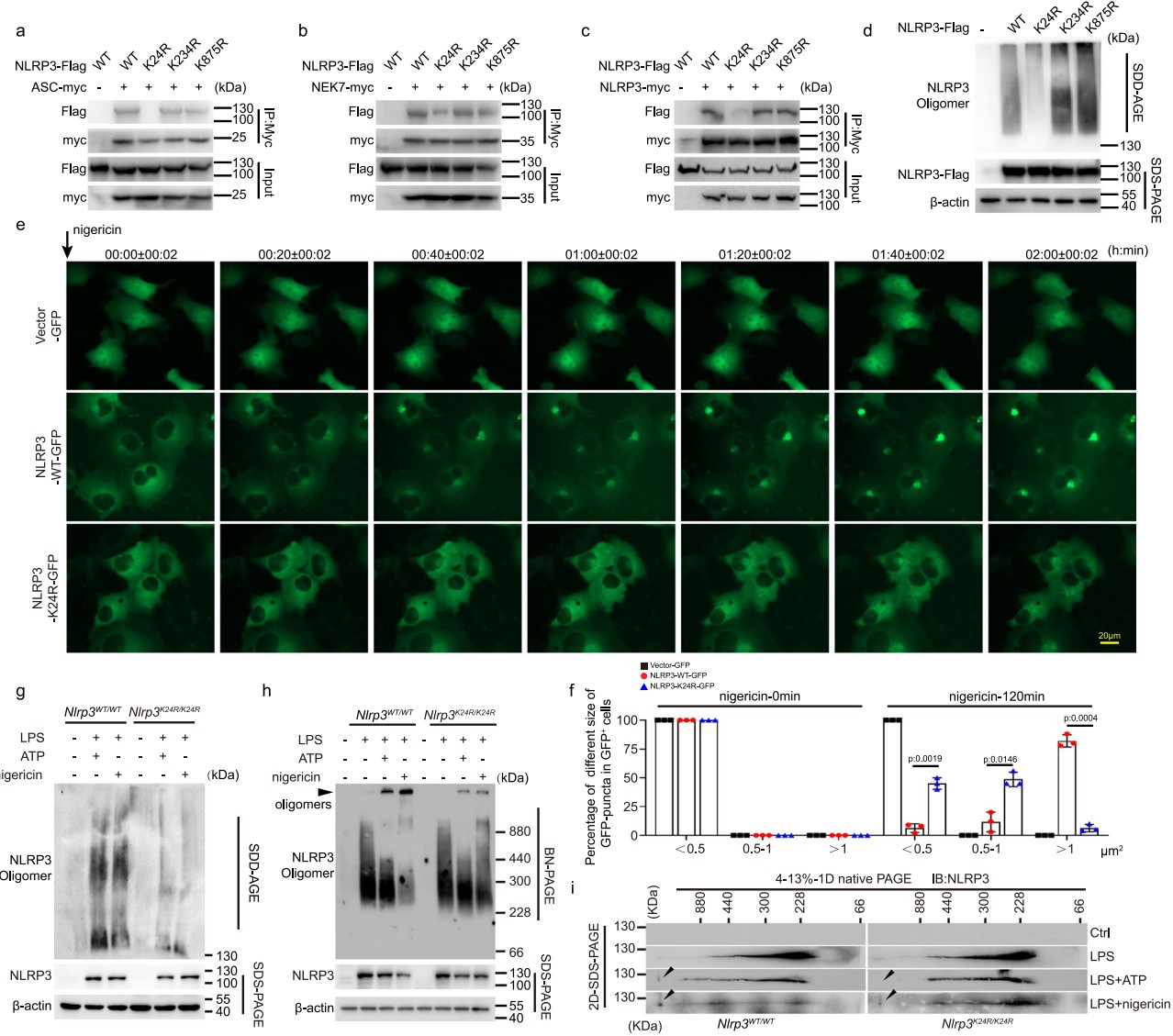

**Fig. 3 | NLRP3 K24 acetylation promotes the inflammasome assembly. a–c** Co-IP analysis of interaction between NLRP3 with ASC, NEK7, or NLRP3 from HEK293T cells transfected with myc-tagged ASC, NEK7, or NLRP3 together with Flag-tagged WT NLRP3 or its nonacetylation mimetic (K-to-R) mutations. **d** Immunoblot analysis of NLRP3 oligomerization and expression by SDD-AGE and SDS-PAGE assay in HEK293T cells transfected with Flag-tagged WT NLRP3 or its nonacetylation mimetic (K-to-R) mutations plasmid and then stimulated with nigericin (10 μM, 1 h). **e, f** COS-7 cells transfected with Vector-GFP, NLRP3 (WT or K24R)-GFP plasmids and stimulated with nigericin (10 μM, 2 h). **e** Representative fluorescent microscopy living-cell images of NLRP3 oligomerization. Scale bar = 20 μm. **f** Ratio of different size of GFP puncta in GFP⁺ cells. $n = 3$ biologically independent experiments. For videos of three representative cells, see Supplementary movies 1, 2 and 3. **g–i** SDD-AGE, BN-PAGE and 2D-SDS-PAGE analysis of NLRP3 oligomers of BMDMs from *Nlrp3^{WT/WT}* or *Nlrp3^{K24R/K24R}* mice treated with LPS (100 ng/mL, 3 h) along or with ATP (5 mM, 1 h) or nigericin (10 μM, 1 h). Results are represented as mean ± SD and typical photographs are representative of three biological independent experiments with similar results. Statistical analyses were carried out via two-way ANOVA with the Bonferroni test for (**f**). Source data are provided as a Source Data file.

macrophages (Supplementary Fig. 9a–c). The suppressive effect was dependent on the acetylation of NLRP3, since NU 9056 declined the acetylation and oligomerization of NLRP3 (Supplementary Fig. 9d–g). To evaluate the inhibitory effects of NU 9056 in vivo, LPS-induced endotoxin shock model and MSU-induced peritonitis model were used. In the endotoxin shock model, mice pre-treated with NU 9056 exhibited markedly reduced IL-1β secretion but not TNF-α production, as compared to the saline pre-treated group (Fig. 7a, b), and the lung tissue also showed less severe tissue damage and diffuse inflammation injury (Fig. 7c, d). Active caspase-1 and mature IL-1β in lungs were decreased with NU 9056 pre-treatment (Fig. 7e). Accordingly, mice pre-treated with NU 9056 were refractory to MSU-induced peritonitis, and exhibited reduced peritoneal neutrophils relative to saline pre-treated mice (Fig. 7f, g, Supplementary Fig. 8). These findings indicated that targeting KAT5-mediated NLRP3 acetylation might provide a new approach for treatment of NLRP3-related diseases.

## Discussion
Several PTMs of NLRP3 have been identified for modulating the activation of the inflammasome, including ubiquitination, phosphorylation, sumoylation and palmitoylation[4,5,8]. However, most of the these PTMs occurs at the priming stage, the PTMs that specifically undergoes at the activation stage remain largely unknown[4,8]. In this study, we demonstrated that NLRP3 is acetylated at Lys24 during the activation stage induced by distinct stimuli, which are required for the oligomerization of NLRP3 and associated assembly of the inflammasome.

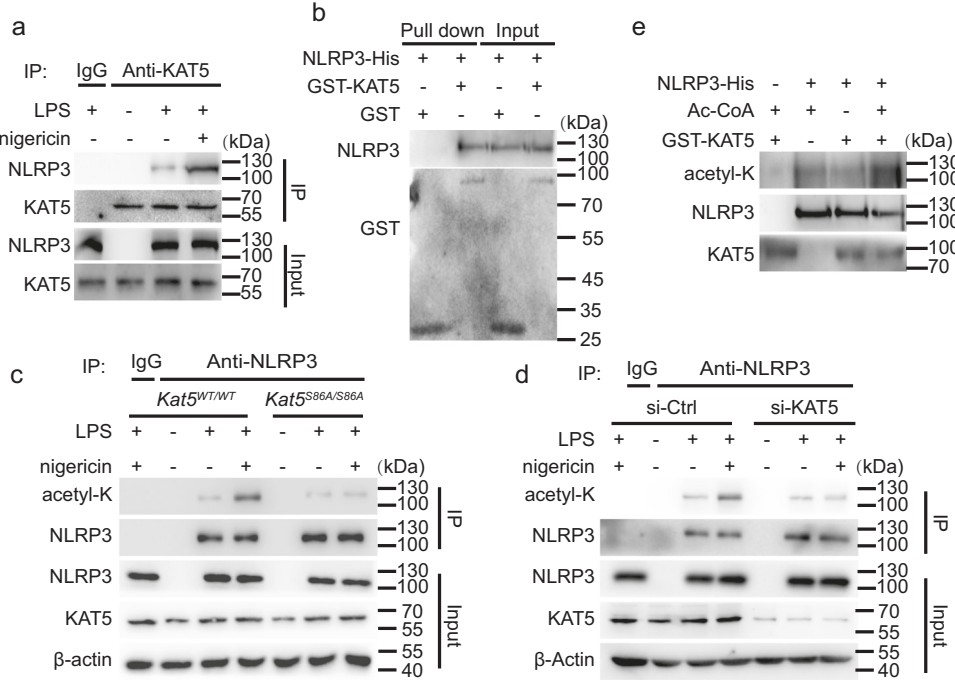

**Fig. 4 | KAT5 mediates NLRP3 acetylation. a** Co-IP analysis of interaction between the endogenous NLRP3 or KAT5 in peritoneal macrophages treated with LPS (100 ng/mL, 3 h) along or with nigericin (10 μM, 1 h). **b** GST pull-down analysis of GST-KAT5 and His-NLRP3. **c** Immunoblot analysis of acetylation level of NLRP3 from *Kat5*^WT/WT^ or *Kat5*^S86A/S86A^ peritoneal macrophage treated with LPS (100 ng/mL, 3 h) along or with nigericin (10 μM, 1 h). **d** Immunoblot analysis of acetylation

level of NLRP3 from peritoneal macrophage transfected with Ctrl or KAT5 siRNA treated with LPS (100 ng/mL, 3 h) along or with nigericin (10 μM, 1 h). **e** Immunoblot analysis of acetylation level of NLRP3 in vitro. The recombinant His-NLRP3 was incubated with GST-KAT5 in the presence or absence of acetyl-CoA (Ac-CoA). Typical photographs are representative of three biological independent experiments with similar results. Source data are provided as a Source Data file.

Moreover, we identified KAT5 mediates NLRP3 acetylation in vitro and in vivo. Thus, our study uncovers that acetylation of NLRP3 is critical for the inflammasome activation.

While this manuscript is in the version of preprint, a study by He et al. reported that sirtuin 2 (SIRT2) deacetylates NLRP3 and inactivates the NLRP3 inflammasome[11] in macrophages. Consistent with our study, they have shown that acetylation of NLRP3 is required to the assembly of the inflammasome. The two complementary researches identified KAT5 and SIRT2 as a writer or eraser for NLRP3 acetylation in macrophages, respectively. In contrast, this study reported that lysine 21 and lysine 22, but not lysine 24, are acetylated during the inflammasome activation. However, we noticed that they used the 3 sites deacetylated mutant (K21/22/24 R) NLRP3 to achieve the best inhibition of the inflammasome in vivo, not excluding the role of acetylation at lysine 24 in the activation of NLRP3. In our study, we generated a polyclonal antibody against NLRP3 acetylated at Lys24 without recognition of NLRP3 acetylated at Lys21 or Lys22(Supplementary Fig. 1d). By using this antibody, we demonstrated that NLRP3 in murine primary macrophages is acetylated at Lys24 upon stimulation by diverse NLRP3 agonists, but not the AIM2 or NLRC4 inflammasome agonists (Fig. 1g, h). Additionally, by generating *Nlrp3*^K24R/K24R^ mice, we demonstrated that K24R mutation indeed impairs NLRP3 inflammasome activation in vitro and in vivo, while this mutation has no effect on NLRC4 or AIM2 inflammasome activation. (Fig. 2). Moreover, we explored the role of acetylation at lysine 24 in the activation of NLRP3 by different means, including immunoprecipitation, SDD-AGE, BN-PAGE and live cell imaging all of which proved the acetylation of NLRP3 at lysine 24 is required for its oligomerization and subsequent activation. Thus, together with He et al.'s study, these data revealed that acetylation of NLRP3 is critical for the NLRP3 inflammasome activation.

We further explored how acetylation affects the function of NLRP3. Since previous studies have shown that PTMs of NLRP3 regulate its

expression, location and association with itself or other inflammasome components[4,5]. By examination all of this events, we observed acetylation of NLRP3 mainly determines its oligomerization and interaction with ASC and NEK7. Recent work suggests that the formation of dispersed TGN and the recruitment of NLRP3 to dTGN are the more earlier events in the activation of the NLRP3 inflammasome[25]. However, acetylation of NLRP3 is not involved in such process, suggesting that acetylation may occurred after the recruitment of NLRP3 to dTGN. Stutz et al. reported that phosphorylation of Ser3 in the PYD domain of NLRP3 disturb the interaction between NLRP3 and ASC by introducing negative charge[31]. Consistently, acetylation of the lysine side chain can neutralize the positive charge, we speculated that acetylation of Lys24 might affect charge-charge interaction for NLRP3 oligomerization. The details could be revealed by the more information on the structural arrangement of NLRP3 activation. Interestingly, Lys24 is evolutionarily conserved across many species (in human is lys26), there is still no related report on this site mutation in human diseases, owing to the loss-of-function mutation is less common than the gain-of-function mutation in NLRP3 related diseases, which needs further investigation.

KAT5 (also known as Tip60), belongs to MYST family of histone acetyltransferase, plays a pivotal role in diverse cellular activities including chromatin remodeling, DNA repair, gene transcription, apoptosis, and tumorigenesis[32,33]. Previous studies have suggested that KAT5 contributes to chronic inflammatory responses[34,35], such as rheumatoid arthritis (RA) and allergy, by regulating the Foxp3 expression in regulatory T cells and STAT6 expression in B cells, respectively[34–36]. Recently, Song et al. demonstrated that KAT5 acetylates cGAS, the DNA sensor for detecting DNA virus, and promotes the innate antiviral responses, giving a good example that acetylation is involved in the regulation of important innate immune sensors[37]. In this study, we revealed that KAT5 acetylates NLRP3 and promotes its oligomerization, and thereby contributes to the activation of the NLRP3 inflammasome. By employing immunoprecipitation, confocal

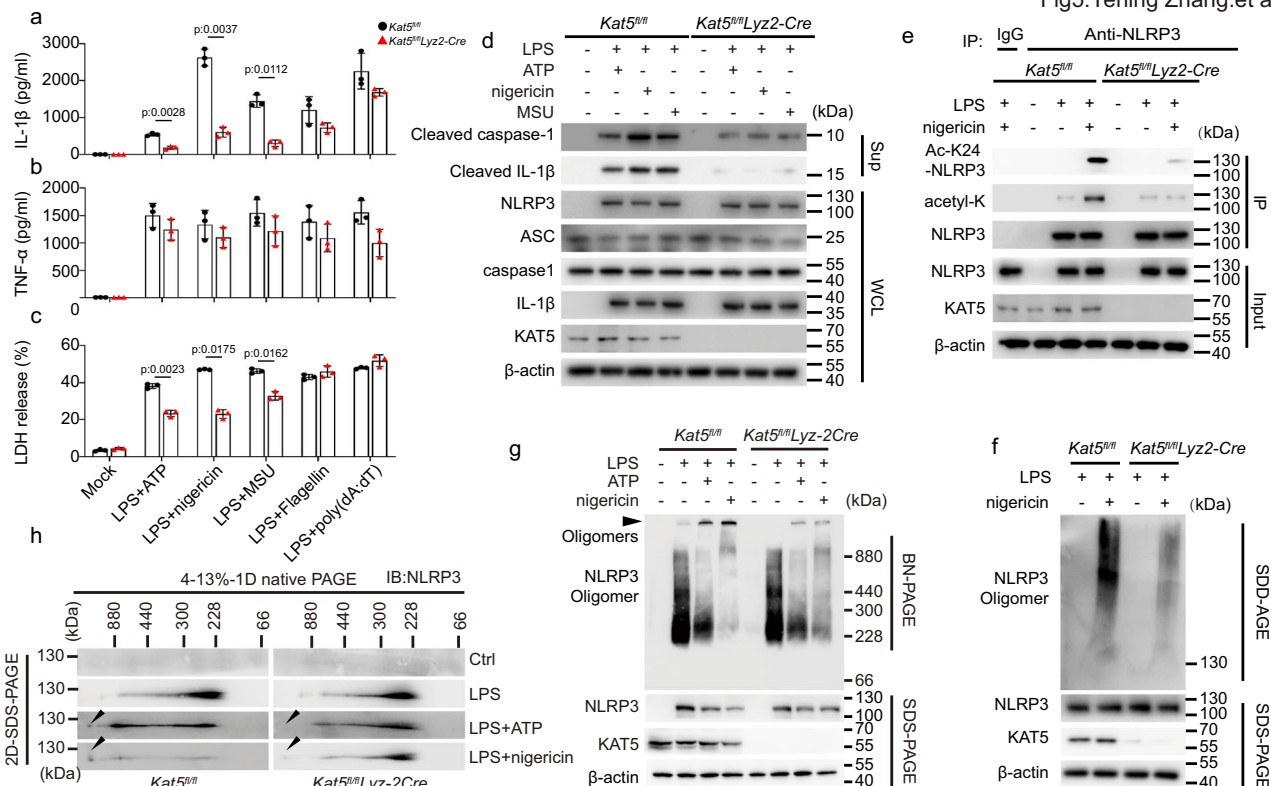

**Fig. 5 | KAT5 specifically promotes NLRP3 inflammasome activation in vitro.**
**a**–**h** BMDMs from *Kat5^fl/fl* (*n* = 3) or *Kat5^fl/fl Lyz2-Cre* (*n* = 3) mice were treated with LPS (100 ng/mL, 3 h) along or with ATP (5 mM, 1 h), nigericin (10 µM, 1 h), MSU (200 µg/ mL, 6 h), Flagellin transfection (2 µg/mL, 1 h) or poly (dA:dT) transfection (1 µg/mL, 16 h). **a**, **b** ELISA analysis of IL-1β and TNF-α in supernatants. **c** Release of LDH in supernatants. **d** Cell Lysates and supernatant were subjected to western blot ana- lysis. **e** Immunoblot analysis of total and K24 acetylation level of NLRP3.

**f**–**h** Immunoblot analysis of NLRP3 oligomerization and NLRP3, KAT5 expression by SDD-AGE, BN-PAGE and 2D-SDS-PAGE assay. Results are represented as mean ± SD and typical photographs are representative of three biological independent experiments with similar results. Statistical analyses were carried out via two-way ANOVA with the Bonferroni test for (**a**–**c**). Source data are provided as a Source Data file.

microscopy, GST pull-down assay and acetylation assay, we proved that KAT5 directly interacts with NLRP3 and induces the acetylation of NLRP3. To further understand the function of KAT5 in the NLRP3 inflammasome activation in vivo, *Kat5^fl/fl lyz2-Cre* mice with KAT5 deletion in myeloid cells and *Kat5^SA/SA* mice with impairment of KAT5 acetyltransferase activity were used. Macrophages from these mice exhibited decreased IL-1β but not TNF-α production in NLRP3 inflam- masome agonists, but not AIM2 or NLRC4 inflammasome agonists, compared to their littermates. Consistently, KAT5 deficiency attenu- ates the systemic inflammatory responses in LPS-induced endotoxin shock model and MSU-induced peritonitis model. Since aberrant NLRP3 inflammasome activity contributes to series of diseases including sepsis, gout, atherosclerosis, type 2 diabetes, autoimmune disorders and Alzheimer's disease[3,38], our study also indicated that targeting KAT5-NLRP3 axis may provide a potential for treatment of NLRP3 associated diseases. Depart from KAT5, KAT2B was also reported to mediate NLRP3 acetylation in prostate cells[39], indicating that the redundancy of histone acetyltransferase in inducing NLRP3 acetylation. Whether exist other acetyltransferase in mediating NLRP3 acetylation in different cell types is worth further investigation.

In summary, our study uncovers that acetylation at lysine 24 by KAT5 is critical for the full activation of NLRP3 inflammasome.

## Methods
### Animal studies
Wild-type C57BL/6 J mice 6–8 weeks old were purchased from Hunan SJA Laboratory Animal Co. Ltd (Changsha, China) (transferred from

National Rodent Laboratory Animal Resources center). Lyz2-Cre mice (Jackson Labs, stock no. 004781) were purchased from Jackson laboratories. *Kat5^fl/fl* and *Kat5^S86A/S86A* mice were gifts from Professor Deepak Bararia[30] and Professor Shengcai Lin[28], respectively. *Nlrp3^K24R/K24R* mice were generated by CRISPR/Cas9-mediated genome editing technology. sgRNA (5'-AATGCATTTTGAATTTCTTG-3') target- ing to the Nlrp3 genome locus was inserted into the sgRNA cloning plasmid. sgRNA and Cas9 mRNA were transcript, modified and purified in vitro using specific kits. The oligo donor (5'-GAGTGTCCGTTGCAAG CTGGCTCAGTATCTAGAGGACCTTGAAGATGTGGATCTTAAGAAATT CAGAATGCATTTGGAAGATTACCCGCCCGAGAAAGGCTGTATCCCAG TCCCCAGGGGCCAGATGGAG-3') was synthesized by Sagan corpora- tion (Shanghai, China). To obtain Nlrp3-mutated mice, the Nlrp3 sgRNA, Cas9 mRNA and oligo donor were co-injected into C57BL6/J fertilized eggs. Pups were genotyped by PCR followed by sequence analysis using primers (mNlrp3-F: 5'-ATGGGGTTTT- CATTCCTGCAC-3'; mNlrp3-R: 5'-GGTTGCTAGGAGATGGGGTT-3'). The found F0 male mice carrying the correct mutation were back-crossed with C57BL/6 J wild-type female mice at least three generations. Het- erozygous mice were then subjected to mate to generate experimental homozygotic *Nlrp3^K24R/K24R* mice and *Nlrp3^+/+* pups. All animals were housed in a 12-h dark/light cycle (25 ± 2°C) under specific pathogen- free (SPF) conditions with unrestricted access to food and water, the experimental and control animals were co-housed. Animal experiments were conducted in accordance with the Institutional Animal Care and Use Committee of Central South University (NO.2018sydw0344).

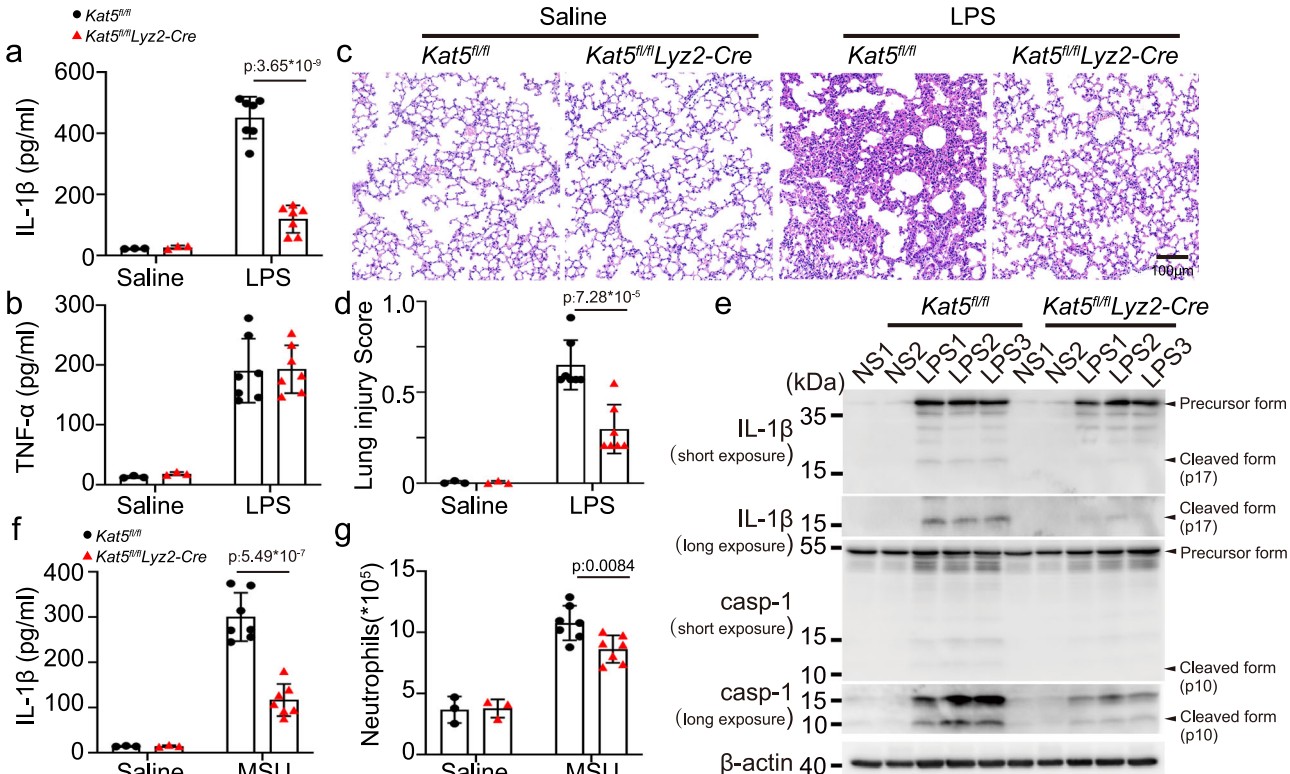

**Fig. 6 | KAT5 promotes NLRP3 inflammasome activation in vivo. a–e** *Kat5*[fl/fl] or *Kat5*[fl/fl]*Lyz2-Cre* male mice of 6–8 weeks were injected with Saline (*n* = 3 biologically independent mice) or LPS (20 mg/kg, i.p.) (*n* = 7 biologically independent mice) for 12 h. **a**, **b** ELISA analysis of IL-1β (p:3.65*10⁻⁹) and TNF-α in serum. **c** representative H&E images of lung sections Scale bar = 100 μm. **d** lung injury score(p:7.28*10⁻⁵). **e** Immunoblot analysis of IL-1β and caspase-1 from lung of *Kat5*[fl/fl] or *Kat5*[fl/fl]*Lyz2-Cre* mice. casp-1(caspase-1) (**f**, **g**) *Kat5*[fl/fl] or *Kat5*[fl/fl]*Lyz2-Cre* male mice of 6–8 weeks were

injected with Saline (*n* = 3 biologically independent mice) or MSU (1 mg, i.p.) (*n* = 7 biologically independent mice) for 8 h. ELISA analysis of IL-1β(p:5.49*10⁻⁷) and quantification of peritoneal exudate neutrophil cells in the peritoneal cavity fluid. Results are represented as mean ± SD. Statistical analyses were carried out via two-way ANOVA with the Bonferroni test for (**a**, **b**–**d**–**f**, **g**). Source data are provided as a Source Data file.

## Endotoxemia

Endotoxemia was induced by injection of LPS (20 mg/kg, i.p.) to male mice of 8 weeks for 12 h, then the mice were deeply anesthetized with pentobarbital (70 mg/kg, ip), the serum and lung tissues were collected for further experiments. To test the effects of KAT5 and NLRP3 K24R mutation on Endotoxemia, littermates of Male mice (*Kat5*[f/f] *Lyz2-Cre* and *Kat5*[f/f], *Nlrp3*[WT/WT] and *Nlrp3*[K24R/K24R]) were used. To test the effects of NU 9056 on Endotoxemia, indicated doses of NU 9056 and equal volume of control solvent were intraperitoneally injected into Wild-Type C57BL/6 J mice 30 min before LPS injection.

## Assessment of histological lung injury

The left lungs were processed for hematoxylin and eosin staining(H&E). Histological examinations were carried out by two experienced researchers in a blinded fashion. Lung injury scores were assessed according to the workshop report of the American Thoracic Society on the features and measurements of experimental acute lung injury in animals[14]. Two experienced researchers were participated in a blinded fashion, 20 random high-power fields (x400 magnification) were scored for each animal.

## MSU-induced peritonitis

For MSU-induced peritonitis, male mice of 6–8 weeks were intraperitoneally injected with 1 mg MSU (dissolved in 500 μL PBS) for 8 h. Mice were euthanized by cervical dislocation. The peritoneal lavage fluids were then collected followed with 500 g for 5 min at 4°C, after that the supernatant were concentrated for ELISA analysis with Amicon Ultra 10 K filter (UFC900308, Millipore). Peritoneal exudate cells were resuspended and counted under microscope, and the

quantity of the neutrophils by FACS. To test the effects of KAT5 on Endotoxemia, littermates of Male mice (*Kat5*[f/f] *Lyz2-Cre* and *Kat5*[f/f]) were used. To test the effects of NU 9056 on Endotoxemia, indicated doses of NU 9056 and equal volume of control solution were intraperitoneally injected into Wild-Type C57BL/6 J mice 30 min before MSU injection.

## Cell lines and culture

**Cell culture.** Primary peritoneal macrophages were generated by intraperitoneally injection of 4 mL 3% thioglycolate into Male mice of 6 - 8 weeks. Three days later, the peritoneal macrophages were collected and maintained in 1640 supplemented with 10% FBS. To obtain Mouse bone marrow-derived macrophages (BMDMs), bone marrow stem cells were first isolated by bone marrow flush and depleted of erythrocytes by hypotonic lyses. After that, bone marrow stem cells were cultured DMEM in the presence of M-CSF (50 ng/ml) and 10% FBS for 5 days to differentiate into BMDM. HEK293T cells and COS-7 cells were obtained from American Type Culture Collection (Manassas, VA), and iBMDMS were kind gifts from Dr. Feng Shao. Those cell lines were cultured in DMEM supplemented with 10% 10% FBS. All the medium contains 100 U/ml penicillin and 100 μg/ml streptomycin.

**Macrophage stimulation.** To induce Inflammasome activation, primary peritoneal macrophages and BMDMs were primed with LPS (100 ng/ml) for 3 h followed by stimulation as follows: ATP (5 mM, 1 h), nigericin (10 μM, 1 h), MSU (200 μg/mL, 6 h), Imiquimod (40 μM 1 h), transfection of Flagellin (2 μg/mL, 1 h) or poly (dA:dT) (1 μg/mL, 16 h) by Lipofectamine 3000. iBMDMs were primed with LPS (1 μg/mL) for 6 h followed by stimulation as follows: ATP (10 mM, 1 h), nigericin

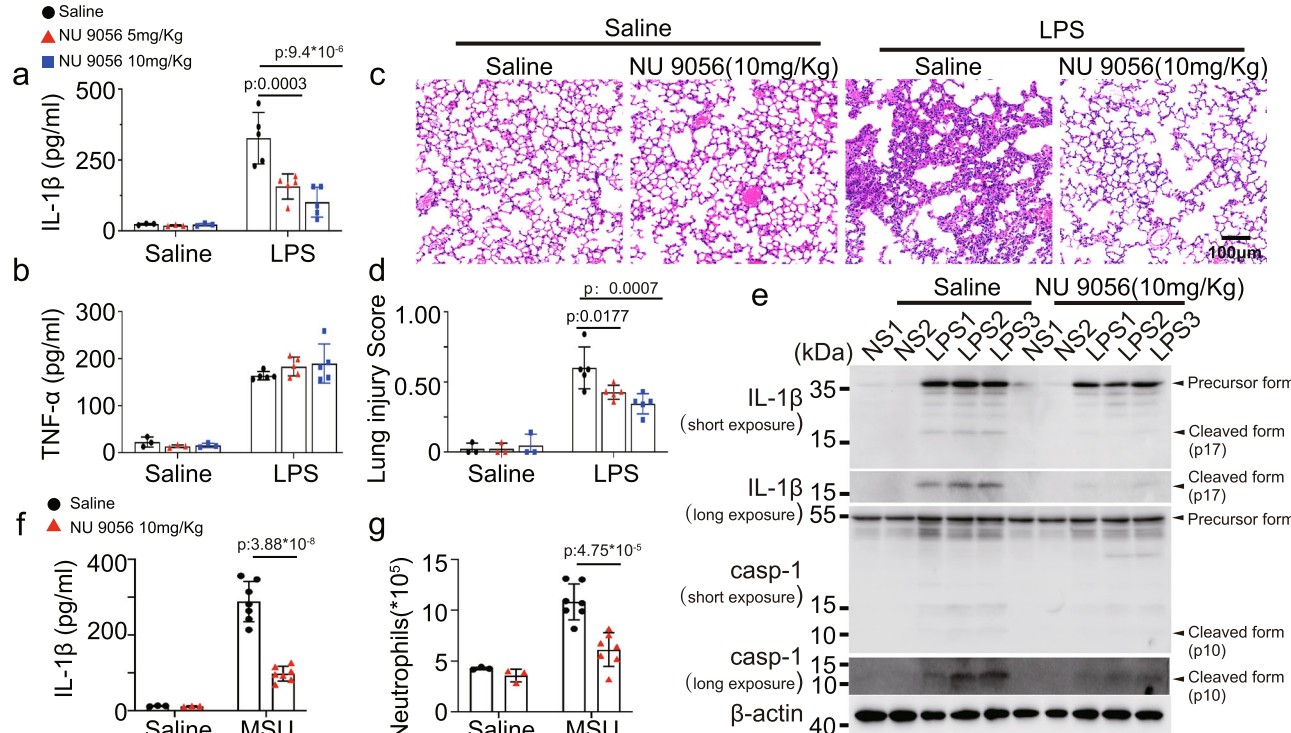

**Fig. 7 | KAT5 inhibitor-NU 9056 suppresses NLRP3 inflammasome in vivo.**
**a**–**e** Wild-type C57BL/6 J male mice of 6-8 weeks were injected with Saline or NU 9056 (5 or 10 mg/kg) 30 min followed with injection of Saline ($n = 3$ biologically independent mice) or LPS (20 mg/kg, i.p.) ($n = 5$ biologically independent mice) for 12 h. **a**, **b** ELISA analysis of serum IL-1β (Saline group VS NU 9056 group p:9.4*10$^{-6}$) and TNF-α. **c** representative H&E images of lung sections. Scale bar = 100 μm. **d** lung injury score. **e** Immunoblot analysis of IL-1β and caspase-1 from lung. casp-1 (caspase-1) (**f**, **g**) Wild-type C57BL/6 J male mice of 6–8 weeks were injected with

Saline or NU 9056 (10 mg/kg) 30 min followed with injection of Saline ($n = 3$ biologically independent mice) or MSU (1 mg, i.p.) ($n = 7$ biologically independent mice) for 8 h. ELISA analysis of IL-1β (p:3.88*10$^{-8}$) and quantification of peritoneal exudate neutrophil cells (p:4.75*10$^{-5}$) in the peritoneal cavity fluid. Results are represented as mean ± SD. Statistical analyses were carried out via two-way ANOVA with the Bonferroni test for (**a**, **b**, **d**, **f**, **g**) Source data are provided as a Source Data file.

---

(20 μM, 1 h), MSU (400 μg/mL, 6 h), transfection of Flagellin (4 μg/mL, 1 h) or poly (dA:dT) (2 μg/mL, 16 h) by Lipofectamine 3000.

**Reconstitution of NLRP3 inflammasome and stimulation in HEK293T cells.** A standard reconstitution system in HEK293T cells was referred[12]. In brief, HEK293T cells were seeded into 24-well plates ($2 \times 10^5$ cells/well) in complete cell culture medium overnight before transfection. After that, the cells were transfected with pro-IL-1β-myc (200 ng/well), pro-caspase-1-myc (100 ng/well), ASC-myc (20 ng/well), NLRP3-Flag (200 ng/well) using Linear Polyethylenimine. 24–36 h later, the culture medium was replaced with 250 μL DMEM cell culture medium, then 10 μM nigericin was added for 1 h. The IL-1β in the supernatant was assessed by ELISA Kit.

Supernatants and cell lysates were collected for Elisa, LDH release and immunoblot analysis

**Identification of NLRP3 acetylation by Mass spectrometry.** HEK293T cells were transfected with Flag-tagged NLRP3 treated with or without nigericin (10 μM, 1 h). Then the cells were lysed in IP buffer (1% (v/v) Nonidet P-40, 50 mM Tris-HCl (pH 7.4), 50 mM EDTA, 150 mM NaCl) contained with a protease inhibitor cocktail. Cell extracts were centrifuged at 12,000 × g for 10 min at 4°C and the supernatants were immunoprecipitated with anti-Flag M2 beads (Sigma), and then dissolved in SDS loading buffer. Samples were denatured at 95°C for 10 min before SDS-PAGE. NLRP3 in gel was digested with trypsin, and subjected to LC/tandem MS (MS/MS) analysis in Jingjie PTM Biolab Co. Ltd. (Hangzhou,China). Briefly, the peptides were analyzed using Q Exactive™ Plus mass spectrometry (Thermo) coupled to an ekspert EASY-nLC 1000 (Thermo). The resulting MS/MS data were processed

using Proteome Discoverer 1.3. Tandem mass spectra were searched against the protein sequence of NLRP3. Mass error was set to 10 ppm for precursor ions and 0.02 Da for fragmentations. Carbamidomethyl on Cys were specified as fixed modification and oxidation on Met and acetylation were specified as variable modifications. Peptide confidence was set at high, and peptide ion score was set >20.

**Plasmids and transfection.** NLRP3, caspase-1, pro-IL-1β, ASC were described as earlier[40], KAT5 full-length sequences were obtained from mouse peritoneal macrophage cDNA, then cloned into pcDNA3.1 vector that contained different tags. Deleted, truncated, and point mutants were generated by PCR-based amplification and the construct encoding the wild-type protein as the template. All constructs were confirmed by DNA sequencing. The primers were as follows: KAT5 forward, 5′-AACGGGCCCTCTAGACTCGAGATGGCGGAGGTGGGGGAG ATAATCGAGGGCT-3′, KAT5 reverse, 5′-TAGTCCAGTGTGGTGGAATT CCCACTTTCCTCTCTTGCTCCAGTCTTTGGGA-3′. NLRP3-K24R forward: 5′-GACCTCAAGAAATTCAGAATGCATTTGGAAGAT-3′ NLRP3-K24R reverse, 5′-ATCTTCCAAATGCATTCTGAATTTCTTGAGGTC-3′. NLRP3-K234R: forward:5′-ACCATCCTAGCCAGGAGGATTATGTTGGA CTGG-3′ NLRP3-K234R reverse, 5′-CCAGTCCAACATAATCCTCCTG GCTAGGATGGT-3′. NLRP3-K875R: forward:5′-CAAGTTTTGTGTGAAA GGATGAAGGACCCACAG-3′ NLRP3-K875R reverse, 5′-CTGTGGGTC CTTCATCCTTTCACACAAAACTTG-3′. Plasmids were transiently transfected into HEK293T cells with Linear Polyethylenimine.

For constructing NLRP3(WT/K24R)-GFP plasmid, NLRP3-WT or NLRP3-K24R full-length sequences were cloned into pEGFP-N1vector. Plasmids were transiently transfected into COS-7 cells with Lipofectamine 3000.

**NLRP3 reconstitution in NLRP3−/− iBMDM cells.** For reconstitution, NLRP3−/− iBMDM cells[40] were transduced with virus stocks containing either a wild-type or K24R mutant NLRP3-encoding lentivirus. Virus was produced in HEK293T cells by co-transfection with pCDH-MCS-EF1-copGFP-NLRP3(wild type or mutants), pSPAX2, and pVSV-G with a ratio of 3:2:1. Virus-containing supernatants were filtered through a 0.45-µm-pore-size filter (Millipore) and supplemented with polybrene (8 µg/ mL) before adding to cells. GFP-positive cells were then sorted by flow cytometry (FACS Aria II, BD Biosciences).

**Knock down of KAT5 in macrophages and iBMDMs.** For peritoneal macrophages, siRNA was transfected using Lipofectamine™ RNAiMAX (Thermo Fisher Scientific) according to the manufacturer's instructions. The siRNA sequences: mouse KAT5 (5′-CCACACUGCAGUAUC UCAATT-3′), and the negative control (5′-UUCUCCGAACGUGUC ACGUTT-3′) were chemically synthesized by Sangon Biotech Co., Shanghai, China. For iBMDMs, shRNA targeting KAT5 were from Genechem Co., Shanghai, China, the sequences were as follows: shKAT5-1(5′-CTGCAACGCCACTTGACCAAA-3′), shKAT5-2 (5′-CTGCTTATTG AGTTCAGCTAT-3′), the negative control(5′-TTCTCCGAACGTGTC ACGT-3′). iBMDMs were transduced with lentivirus-KAT5-RNAi or ctrl virus. Fourty-eight hours later, the cells were selected by culture with 5 mg/mL puromycin. Single colonies were obtained by serial dilution and amplification.

**Immunofluorescence.** For immunostaining, cells were fixed with 4% paraformaldehyde and permeabilized with 0.1% Triton X-100 in PBS, then the cells were incubated with primary antibodies(anti-NLRP3 antibody (1:100), anti-KAT5 antibody (1:100), anti-GOLGA4(1:200) or anti-TGN38(1:200)) followed by staining with secondary antibody(DyLight 488-labeled secondary antibody (Invitrogen) (1:50),Alexa Fluor 594-conjugated secondary Ab (Invitrogen) (1:50),Alexa Fluor® 488 Goat anti-mouse IgG (minimal x-reactivity) Antibody(1:100) or Cy3−conjugated Affinipure Goat Anti-Rabbit IgG(H + L) (1:100)), while nuclei were stained with DAPI containing mounting medium (Beyotime). To better preserve the dTGN structures in COS-7 cells, 0.01% Triton X-100 and 0.1% saponin were respectively in place of 0.1% Triton X-100 in permeabilization step for immunostaining of GOLGA4 and TGN38. Fluorescence images for fixed cells were taken with confocal fluorescence microscope (SpinSR10; Olympus) and (LSM800; ZEISS).

**Living cell imaging.** Live cell imaging was performed by ZEISS Axio observer7 equipped with an incubator, allowing live cell imaging at 37 °C and at 5% CO2. Before Live cell imaging, COS-7 Cells were transfected with indicated plasmids using Lipofectamine 3000. Thirty-six hours later, cells were digested and reseeded into NEST glass bottom cell culture dish (801001) at density of $2 \times 10^5$ cells overnight. Imaging started immediately after Nigericin stimulation and lasted for 2 h. The immunofluorescence images were statistically analyzed by Zen 3.6 and Fiji image J (Version 2.9.0)

**Quantitative PCR.** Total RNA was extracted by using RNA Fast 200 kit according to the manufacturer's instructions (FASTAGEN). Complementary DNA was synthesized by using TransScript All-in-One First-Strand cDNA Synthesis SuperMix for qPCR (TransGen Biotech) according to the manufacturer's protocols. Quantitative PCR was performed using SYBR Green (Vazyme Biotech) on a LightCycler 480 (Roche Diagnostics), and data were normalized to β-actin expression. The $2^{-\Delta\Delta CT}$ method was used to calculate relative expression changes. Gene-specific primers were as follows: KAT5 forward, 5′-TCCCG GTCCAGATCACACTC-3′; KAT5 reverse, 5′-ACCTTCCGTTTCGTTG AGCG-3′; NLRP3 forward, 5′-TGGATGGGTTTGCTGGGGAT-3′, NLRP3 reverse, 5′-CTGCGTGTAGCGACTGTTGAG-3′; IL-1β forward, 5′- GCAA CTGTTCCTGAACTCAACT-3′ IL-1β reverse, 5′- ATCTTTTGGGGTCC

GTCAACT-3′; Caspasse-1 forward, 5′-ACAAGGCACGGG ACCTATG-3′; Caspasse-1 reverse, 5′-TCCCAGTCAGTCCTGGAAATG-3′.ASC forward, 5′-CTTGTCAGGGGATGAACTCAAAATT-3′; ASC reverse, 5′-GCCAT ACGACTCCAGATAGTAGC-3′. β-actin forward, 5′-AGTGTGACGTTGAC ATCCGT-3′; β-actin reverse, 5′-GCAGCTCAGTAACAGTCCGC-3′

**Immunoprecipitation and Western blot.** To detect NLRP3 acetylation, cells were lysed in immunoprecipitation (IP) buffer supplemented with 0.1 mM PMSF and EDTA-free protease inhibitor cocktail, 1 µM trichostatin A (TSA) and 10 mM nicotinamide (NAM), followed by sonication and centrifugation at 13523 g for 10 min at 4 °C. The supernatants were immunoprecipitated with NLRP3 antibody for 12 h at 4 °C. the next day, protein A/G plus agarose were added followed by 2 h rotation at 4°C. The immunoprecipitants were washed six times with IP buffer and boiled in 1 × SDS-loading buffer for immunoblot analysis.

To detect protein interactions, macrophages and 293 T cells were lysed in IP buffer supplemented with 0.1 mM PMSF and EDTA-free protease inhibitor cocktail followed with centrifugation at 13523 g for 10 min at 4 °C. The supernatants from macrophage samples were incubated with specific Abs or IgG and rotated at 4 °C overnight, the next day, protein A/G plus agarose were added followed by 2 h rotation at 4 °C. For supernatants from 293 T cells, anti-Flag or anti-Myc affinity gel were added followed with 2 h rotation at 4 °C. After that, the Immunoprecipitants were washed four times with IP buffer and boiled in 1×SDS loading buffer for immunoblot analysis.

For whole cell lysate and histological immunoblot analysis, cells or tissues were lysed with CLB buffer (CST) supplemented with 0.1 mM PMSF and EDTA-free protease inhibitor cocktail, and then protein concentrations in the extracts were measured with a bicinchoninic acid assay (Pierce). Equal amounts of extracts were boiled in 1 × SDS loading buffer for immunoblot analysis.

**GST Pulldown assay.** Recombinant murine NLRP3-His and KAT5-GST proteins were purchased from Sino Biological Inc (Beijing, China), which were expressed by eukaryotic expression system. For GST Pulldown assay, 2 µg KAT5-GST and GST protein were incubated with 10 µL Glutathione Sepharose™ 4B (GE Healthcare) in 500 µL GST pull-down buffer (20 mM HEPES, pH 7.9, 150 mM NaCl, 0.5 mM EDTA, 1 mM DTT, 10%Glycerol, 0.1% Triton-X-100) for 30 min rotation at 4 °C, then the resins were washed thrice with ice-cold PBS at 845 g for 3 min and resuspended in 40 µL GST pull-down buffer. After that immobilized KAT5-GST and GST protein on GSH beads were incubated with 2 µg NLRP3-His for 2 h at 4 °C, after that the Beads were washed thrice with GST pull-down buffer and boiled in 1 × SDS-loading buffer for immunoblot analysis.

**In vitro acetylation assay.** The acetylation assay was performed by incubating 1ug recombinant murine NLRP3-His ± 1ug KAT5-GST protein in 40 µL of reaction buffer(20 mM Tris-HCl (pH 8.0), 20% Glycerol, 100 mM KCl, 0.2 mM EDTA, 1 mM DTT ± 100 µM Acetyl-CoA) according to the previous research[28]. After incubation for 1 h at 30 °C, the reaction was stopped and boiled in 1 × SDS-loading buffer for immunoblot analysis.

**Acety-NLRP3 K24 antibody**
The antibody of Acetyl-NLRP3-K24 was customized produced by ABclonal (Wuhan, China). It was generated by immunizing rabbits with the acetyl-lysine-peptide KF(K-Ac)-Nle-HLED-C, covalently cross-linked to keyhole limpet hemocyanin (KLH). The Acetyl-NLRP3-K24 (1:200 dilution) was utilized for the detection of the K24 acety-NLRP3.

**SDD-AGE.** The oligomerization of NLRP3 was analyzed according to the previous report[20]. Cells were lysed with Triton X-100 lysis buffer (0.5% Triton X-100, 50 mM Tris-HCl, 150 mM NaCl, 10% glycerol, 1 mM

PMSF, and protease inhibitor cocktail), which were then resuspended in $1\times$ sample buffer ($0.5\times$ TBE, 10% glycerol, 2% SDS, and 0.0025% bromophenol blue) and loaded onto a vertical 1.5% agarose gel. After electrophoresis in the running buffer ($1\times$ TBE and 0.1% SDS) for 1 h with a constant voltage of 80 V at 4 °C, the proteins were transferred to Immobilon membrane (Millipore) for immunoblotting. $1\times$ TBE buffer contains 89 mM Tris, pH 8.3, 89 mM boric acid, and 2 mM EDTA.

**BN-PAGE.** Blue native gel electrophoresis was performed according to the previous report[16], cells were lysed with ice-cold Native-PAGE lysis buffer (20 mM Bis-tris, 500 mM-aminocaproic acid, 20 mM NaCl,10% (w/) glycerol, 0.5% digitonin, 0.5 mM Na3VO4, 1 mM PMSF, 0.5 mM NaF, 1x EDTA-free Roche protease inhibitor cocktail, pH 7.0) for 30 min on ice. After centrifugation at 20,000 g for 30 min at 4 °C, supernatants were mixed with Coomassie G-250 to a final concentration of 0.25%, and then equal sample were separated by 4–13% BeyoGel™ Blue Native Precast PAGE Gel (Beyotime). Native gels were soaked in 1% SDS solution for 15 min before being transferred to PVDF membranes (Millipore), followed by conventional western blotting. In 2D SDS-PAGE, a gel slice of the natively resolved gel was placed in a dish containing 1x SDS-PAGE sample buffer (WB-0091) for 10 min, microwaved on high for 20 s, and rocked for another 15 min at room temperature before loading the slice into a well 8% SDS-PAGE gel, followed by conventional western blotting.

**Flow cytometry.** The quantification of peritoneal exudate neutrophil cells was measured by FACS. After counting the number of harvested peritoneal exudate cells, 1.5*10^6 cells were obtained and washed with PBS, then resuspended with 100 µl binding buffer softly; 0.05 µl FITC anti-mouse/human CD11b Antibody and 0.1 µl PE anti-mouse Ly-6C Antibody were added and incubated in a dark room for 30 min. The ratio of neutrophil cells in each group were measured by BD LSRFortessa and analyzed with FlowJo software (FlowJo, LLC). Cells were discriminated from debris and clumps using the FSC-A/SCC-A gating strategy based on experience. Only single cells were used by using an FITC/PE gating strategy and selecting cells along the diagonal.

**ELISA assay for cytokines.** Levels of IL-1β, TNF-α and IL-6 collected from cell culture and sera were determined using quantitative ELISA kits (eBioscience) according to the manufacturer's instructions.

**LDH release assays.** Levels of LDH in supernatant were determined using LDH Cytotoxicity Assay Kit (Beyotime) according to the manufacturer's instructions.

**Statistical and reproducibility.** All statistical analysis was performed with GraphPad Prism 9.4.1. Data are shown as the mean ± standard deviation (SD). one-way analysis of variance (ANOVA) followed with Dunnett's test and two-way ANOVA followed with Bonferroni test was performed. Differences with a $p$-value $< 0.05$ were considered to be statistically. Statistical details of analyses can be found in the figure legends. Mice were randomly divided to different groups in animal studies. No samples or animals were excluded from the analysis. No statistical method was used to predetermine the sample size. All samples were processed in blind fashion. The number of independent samples of each experiment can be found in the relevant figure legend. For western blots, qPCR, flow cytometry and other quantitative experiment were performed at least three times. For IF/IHC/HE stains, the number of independent samples is consistent with the relevant quantitative graph, the images are representative images of at least three independent replicates of the experiments.

**Reporting summary**
Further information on research design is available in the Nature Portfolio Reporting Summary linked to this article.

## Data availability

All data supporting the findings of this study are available within the article and the data generated in this study are provided in the Supplementary Information and Source data file. Source data are provided with this paper. The mass spectrometry data generated in this study have been deposited in the ProteomeXchange Consortium database under accession code PXD041763 Source data are provided with this paper.

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

## Acknowledgements

We thank Dr. Sheng-Cai Lin (Xiamen University) for providing *Kat5^SA/SA* mice, Dr. Susumu Kobayashi (Harvard Medical School) for providing *Kat5^flox* mice and Dr. Feng Shao (National Institute of Biological Sciences) for providing immortalized mouse macrophages. We thank Dr. Jian Qiu (Central South University) for providing technical help and thank for Qianqian Xue and Ling Li for assisting in raising the animals. This work was supported by National Key Research and Development Program of China (grant nos. 2021YFC2500802 to K.Z.), National Natural Science Foundation of China (grant nos.82272207 to K.Z., grant nos.82102281 to N.Z., grant nos.81930059 to B.L.), National Outstanding Youth Science Fund Project of the National Natural Science Foundation (grant nos. 82025021 to B.L.), the Provincial Natural Science Foundation of Hunan in China (grant nos.2021JJ20090 to K.Z., grant nos. 2023JJ40916 to Y.Z.), the Wisdom Accumulation and Talent Cultivation Project of the Third xiangya hospital of Central South University (grant nos. JC202201 to K.Z.), China Postdoctoral Science Foundation (no. 2022M713539 to Y.Z.), Graduate Student Research Innovation Project in Hunan Province (grant nos.CX20230316 to L.L.)

## Author contributions

K.Z. and B.L. supervised the whole project; K.Z. designed the research; K.Z. Y. Z. and B.L. wrote the manuscript; Y.Z. performed the experiments, analyzed the data and made the figures; L.L., X.X. and F.W. performed the experiments; J.W. and Y.L. raised the animals and helped with data analyses and discussions; N.Z. assisted in data interpretation and edited the manuscript; Y.D. provided help for the revision experiments.

## Competing interests

The authors declare no competing interests.
