## [Peer Review File · Nature Communications]

Acetylation is required for full activation of the NLRP3 inflammasomeREVIEWER COMMENTS

Reviewer #1 (Remarks to the Author):

Zhang et al report that acetylation of NLRP3 at Lys24 is required for the NLRP3 inflammasome activation. They also determined that KAT5 acetylates NLRP3 and promotes its oligomerization leading to the activation of the NLRP3 inflammasome. Lastly, they nicely showed that the genetic loss of KAT5 and the pharmacological inhibition of KAT5 by NU9056 allowed to repress the NLRP3 inflammasome activation. This manuscript complements our knowledge of the modulation of NLRP3 activation through the identification of KAT5 as a writer of NLRP3 acetylation. Overall, this body of research is quite nicely done but I do have some concerns listed below:

Minor comments:

- the puncta are not obvious in Figure 2e.
- In Figures 5C and 6C, quantification for the lung injury score should be included.
- What are the inducers of AIM2 and NLRC4 used in the study?
- The number of mice used in Figure 4a,b,c should be indicated.

Reviewer #2 (Remarks to the Author):

Zhang and colleagues suggest that the NLRP3 triggers that can provide signal number 2 (i.e., the step after the priming event) also promote acetylation of NLRP3 at lysine 24 and that this regulates NLRP3 oligomerization and assembly into an active signaling competent structure. The investigators demonstrate that 1. the enzyme KAT5 is responsible for the acetylation of NLRP3 as loss of KAT5 in myeloid cells and pharmacological inhibition of KAT5 led to lower NLRP3 inflammasome activation in cells and in mice.

Overall the data are convincing, and the work moves the field forward. There is a lot of emphasis on positioning KAT5 as a pharmacological target upstream of NLRP3. While this is of potential interest, the authors fail to recognize that these types of enzymes may have multiple targets, and the effects seen could be independent of the acetylation of NLRP3. Hence, the authors need to provide an experiment with the inhibitor in the context of in vivo inflammasome activation of another inflammasome (such as NLRC4, AIM2,...). Such an experiment would establish that the NLRP3 pathway is indeed dependent on acetylation by KAT5 in vivo.

Minor comments:

The Anti-acetyl NLRP3 antibody should be not only tested on the peptides but also on full-length WT NLRP3 protein that lacks the acetylation site. This could quickly be done using IP/WB analysis using their existing cell lines.

Figure 2. The imaging data only show a time course analysis of one cell per condition. This is insufficient to judge the phenotype, as selection bias is a common issue in microscopy. Please provide data from larger fields of view and perform an image analysis with statistics.

Reviewer #3 (Remarks to the Author):

The manuscript suggests that acetylation of NLRP3 at lysine 24 plays a critical role in the oligomerization and assembly of NLRP3. The authors found that this acetylation does not affect the recruitment of NLRP3 to the dispersed trans-Golgi network (TGN). Additionally, the authors identified that KAT5, a specific enzyme, is responsible for mediating the acetylation of NLRP3. While this manuscript presents novel findings on the identification of the acetylation site of NLRP3

and provides evidence through mutation experiments that this site is crucial for inflammasome activation and oligomerization, a significant portion of the results in this paper had already been preprinted on bioRxiv by the authors in 2019, and since then, many papers have cited those results. Based on the subsequent publication of several papers highlighting the significance of NLRP3 acetylation in inflammasome activation, it can be inferred that this manuscript is deemed to have limited novelty and importance. Furthermore, the data and experimental methods presented in this manuscript regarding inflammasome activation and oligomerization are not considered to be of high quality. In addition, a comprehensive exploration of the molecular pathways linked to the inhibition of inflammasome activation resulting from KAT5 gene deletion, particularly in an in vivo model, is required. Consequently, considering the overall insufficiency of the research findings, this manuscript is deemed unsuitable for publication in "Nature Communications."

Major Comments

1. It needs to be demonstrated whether ATP, MSU, Imiquimod, or nigericin alone, without the presence of LPS, induces NLRP3 acetylation.
2. There is no mention of AIM2, NLRC4, flagellin, or poly dA-dT in this manuscript. Furthermore, it is necessary to include information about nigericin in the "Introduction" section of the paper.
3. The presence of the cleaved form of active caspase-1 is a crucial indicator of inflammasome activation. However, the results presented in the manuscript only show the presence of the proform of caspase-1 and not the cleaved form. This discrepancy needs to be clarified and addressed in order to provide a more accurate assessment of inflammasome activation.
4. Histological staining for speck protein components, active caspase-1, or mature IL-1 β is necessary to assess inflammasome activation in situ on the tissue shown in Figures 5 and 6. This staining technique would allow for the visualization and localization of these specific components within the tissue, providing valuable insights into the process of inflammasome activation.
5. The NLRP3 oligomerization assay presented in the manuscript is not clearly demonstrated. As seen in other studies, it is necessary to show a distinct ladder-like pattern indicative of NLRP3 oligomerization.

Minor Comments

1. It appears that caspase-1-myc was transfected; however, it needs to be clarified why caspase-1 is not being expressed in the first lane of Figure 1C.
2. The image quality of Figure 2e is low. Is there a specific reason for the observed differences in cell size? Could it be attributed to potential transfection-related damage? Is there a specific rationale for utilizing the Cos7 cell line, a monkey kidney cell line, instead of the HEK293 cell line?
3. The target protein of the various inhibitors used in Figure S3a should be mentioned. Additionally, it is essential to assess whether the inhibitors effectively performed their intended functions.

Reviewer #4 (Remarks to the Author):

Zhang and co-authors claim that acetylation of NLRP3 at lysine 24 by lysine acetyltransferase 5 (KAT5) promotes its oligomerization and inflammasome assembly. This claim is supported by data from NLRP3-reconstituted 293T cells and immortalized bone marrow-derived macrophages as well as by mouse models of LPS-induced endotoxin shock and peritonitis driven by the NLRP3 inflammasome activator, monosodium urate. Multiple approaches, including unbiased proteomics-based mass spectrometry, genetic and pharmacological manipulations are used to firm up the conclusions. Additional strengths of this manuscript include its concise and clear narrative (though attention to some wordings is needed) and the novelty of the finding as regulation of the NLRP3 inflammasome by acetylation has not been extensively studied. However, there are concerns that need to be addressed.

Major concerns:

- Are NLRC4 and AIM2 acetylated in responses to their respective activators?
- Are blots such as those shown in Fig.2 A-C been cut and pasted or not? Lanes for the different

proteins do not align well.

- Indicate where NLRP3 monomer, dimer, etc..., migrate in Fig. 2D. How do the authors explain the diffuse but not distinctive NLRP3 species (monomers, dimers, etc...)? This concern, which also applies to Fig. 4f and Supplementary Fig. 7d, needs to be convincingly addressed as it cast doubt on the scientific rigor.
- Quantitative data are needed for Supplementary Fig. 2a, b.
- The quality of the images in Fig. 5C and Fig. 6C is not optimal.
- Previous studies on NLRP3 acetylation, for example by KAT2B (PMID: 36250925), are not mentioned.

Minor concerns:

- Several phrases (e.g., 2nd sentence of the results) are too long.
- Explain what Ack means on the blots of Fig. 1.
- Pay attention to wordings such as inapparent, enzymatical, etc...
- IL-1 maturation reflects the activity of the NLRP3 inflammasome, but not NLRP3.

Reviewer #1 (Remarks to the Author):

Zhang et al report that acetylation of NLRP3 at Lys24 is required for the NLRP3 inflammasome activation. They also determined that KAT5 acetylates NLRP3 and promotes its oligomerization leading to the activation of the NLRP3 inflammasome. Lastly, they nicely showed that the genetic loss of KAT5 and the pharmacological inhibition of KAT5 by NU9056 allowed to repress the NLRP3 inflammasome activation. This manuscript complements our knowledge of the modulation of NLRP3 activation through the identification of KAT5 as a writer of NLRP3 acetylation. Overall, this body of research is quite nicely done but I do have some concerns listed below:

Answer: In accordance with these valuable suggestions, we carefully revised the manuscript and performed a number of additional experiments. By addressing these comments and concerns, we have strengthened the conclusion of our paper and the physiological relevance of our findings. The point-by-point responses to comments were listed as below.

Minor comments:

- the puncta are not obvious in Figure 2e.

Answer: We replaced the Figure 2e with new data which exhibited more cells in a view with high quality in the modified version of manuscript. (**Fig 3e in the revised version**).

- In Figures 5C and 6C, quantification for the lung injury score should be included.

Answer: We assessed of the histological lung injury according to the workshop report of the American Thoracic Society on the features and measurements of experimental acute lung injury in animals (**Am J Respir Cell Mol Biol.** 2011;44(5):725-38) in the modified version of manuscript (**Fig 2f.6c.7c in the revised version**).

- What are the inducers of AIM2 and NLRC4 used in the study?

Answer: Poly(dA:dT) and flagellin transfection were used to activate AIM2 and NLRC4 inflammasome, respectively. We added the details at the first paragraph to the "results" section.

- The number of mice used in Figure 4a,b,c should be indicated.

Answer: We have numbered the mice in the figure legend in the modified version of manuscript (**Fig 5a-c in the revised version**).

Reviewer #2 (Remarks to the Author):

Zhang and colleagues suggest that the NLRP3 triggers that can provide signal number 2 (i.e., the step after the priming event) also promote acetylation of NLRP3 at lysine 24 and that this regulates NLRP3 oligomerization and assembly into an active signaling competent structure. The investigators demonstrate that 1. the enzyme KAT5 is responsible for the acetylation of NLRP3 as loss of KAT5 in myeloid cells and pharmacological inhibition of KAT5 led to lower NLRP3 inflammasome activation in cells and in mice.

Answer: In accordance with the valuable suggestions, we carefully revised the manuscript and performed additional experiments. By addressing these comments and concerns, we have strengthened the conclusion of our paper and the physiological relevance of our findings. The point-by-point responses to comments were listed as below.

Overall the data are convincing, and the work moves the field forward. There is a lot of emphasis on positioning KAT5 as a pharmacological target upstream of NLRP3. While this is of potential interest, the authors fail to recognize that these types of enzymes may have multiple targets, and the effects seen could be independent of the acetylation of NLRP3. Hence, the authors need to provide an experiment with the inhibitor in the context of *in vivo* inflammasome activation of another inflammasome (such as NLRC4, AIM2,...). Such an experiment would establish that the NLRP3 pathway is indeed dependent on acetylation by KAT5 *in vivo*.

Answer: As suggested, we examined the effects of NU9056 on HSV-1 infected mice, since HSV-1 can induce the AIM2 inflammasome activation. (**Cell host & microbe**,2018;23(2):254-265.e7; **Nature**,2021;597(7876):415-419.)The results showed that NU9056 had no effect on HSV-1 induced secretion of IL-1 β or TNF- α (**Figure R1**), in accordance with the experiments *in vitro* (**Supplementary Figure 9a-c in the revised version**). Although we got the negative results of NU9056 on AIM2

inflammasome activation *in vivo*, we could not exclude the other targets of KAT5 *in vivo*, such as cGAS (PNAS. 2020;117(35):21568-75.), FOXP3 (Immunity. 2012;36(5):717-30.), and PI3K/AKT signal (Molecular cell. 2018;70(2):197-210 e7.), which may affect inflammatory responses. To further explain that target NLRP3 acetylation could provide potential for treatment associated inflammatory disorders, we generated a knock in (KI) mice which harboring the *Nlrp3* K24R allele (*Nlrp3*^{K24R/K24R}). The expression of NLRP3 was not affected in *Nlrp3*^{K24R/K24R} primary macrophages (Supplementary Figure 2c in the revised version). However, when challenged with ATP, nigericin and MSU, the secretion of IL-1 β , but not TNF- α , was impaired in *Nlrp3*^{K24R/K24R} primary macrophages compared to WT cells (Fig 2a-c in the revised version). We also performed LPS-induced endotoxin shock model in this KI mice and their WT littermate controls. In response to LPS challenge, the serum levels of IL-1 β , but not TNF- α (Fig 2d, e in the revised version) were significantly decreased in *Nlrp3*^{K24R/K24R} compared to WT mice. Moreover, reduced lung injury of *Nlrp3*^{K24R/K24R} mice were observed (Fig 2f-h in the revised version). Thus, these results suggest that targeting KAT5-mediated acetylation of NLRP3 may be implicated in the treatment of such inflammatory diseases, which still needs further investigation.

Figure R1 The effects of NU 9056 in HSV-1 infection *in vivo*

Wild-type C57BL/6 mice (6-8 weeks old) were intraperitoneally injected with Saline or DMSO (10mg/Kg), after 30min, 5×10^7 plaque-forming units (PFU) of HSV1 were injected(i.p.). 24 hours later, the mice were sacrificed, the secretion of IL-1 β and TNF-a in serum were analyzed.

Minor comments:

The Anti-acetyl NLRP3 antibody should be not only tested on the peptides but also on full-length WT NLRP3 protein that lacks the acetylation site. This could quickly be done using IP/WB analysis using their existing cell lines.

Answer: According to the valuable suggestion, we overexpressed WT and K24R NLRP3 in HEK293T cells with or without nigericin treatment, we observed the Anti-acetyl 24 NLRP3 band in WT NLRP3 overexpression group stimulated with nigericin, further demonstrating the specificity of the antibody we produced (**Supplementary FigS1e in the revised version**).

Figure 2. The imaging data only show a time course analysis of one cell per condition. This is insufficient to judge the phenotype, as selection bias is a common issue in microscopy. Please provide data from larger fields of view and perform an image analysis with statistics.

Answer: We accepted the valuable suggestion and replaced the Figure 2e with new data which showed more cells in a view with high quality and also provided the statistics analysis (**Fig3e-f in the revised version**).

Reviewer #3 (Remarks to the Author):

The manuscript suggests that acetylation of NLRP3 at lysine 24 plays a critical role in the oligomerization and assembly of NLRP3. The authors found that this acetylation does not affect the recruitment of NLRP3 to the dispersed trans-Golgi network (TGN). Additionally, the authors identified that KAT5, a specific enzyme, is responsible for mediating the acetylation of NLRP3. While this manuscript presents novel findings on the identification of the acetylation site of NLRP3 and provides evidence through mutation experiments that this site is crucial for inflammasome activation and oligomerization, a significant portion of the results in this paper had already been preprinted on bioRxiv by the authors in 2019, and since then, many papers have cited those results. Based on the subsequent publication of several papers highlighting the significance of NLRP3 acetylation in inflammasome activation, it can be inferred that this manuscript is deemed to have limited novelty and importance. Furthermore, the data and experimental methods presented in this manuscript regarding inflammasome activation and oligomerization are not considered to be of high quality. In addition, a comprehensive exploration of the molecular pathways linked to the inhibition of inflammasome activation resulting from KAT5 gene deletion, particularly in an *in vivo* model, is required. Consequently, considering the overall insufficiency of the research findings, this manuscript is deemed unsuitable for publication in "Nature Communications."

Answer: Although this manuscript had already been preprinted on bioRxiv in 2019, we did not provide tough evidence that KAT5 is responsible for NLRP3 acetylation *in vivo*. In this revised manuscript, we strengthened the conclusion and the physiological relevance of our findings by using *Kat5*-myeloid conditional knock out mice, *Kat5*-S86A mutant mice and *Nlrp3*-K24R mutant mice. Furthermore, we provided high quality images and performed BN-PAGE for detection of NLRP3 oligomerization. The point-by-point responses to comments were listed as below.

Major Comments

1. It needs to be demonstrated whether ATP, MSU, Imiquimod, or nigericin alone, without the presence of LPS, induces NLRP3 acetylation.

Answer: According to the suggestions, we detected the acetylation of NLRP3 in primary macrophages only treated with ATP, MSU, Imiquimod, or nigericin, we did not observe the acetylation band in NLRP3(**Figure R2**).

Figure R1 Analysis of NLRP3 acetylation upon different NLRP3 agonist stimulation alone.

Peritoneal macrophages were treated with or without LPS (100 ng/mL, 3 h) then followed with ATP (5 mM, 1 h), nigericin (10 μ M, 1 h or Imiquimod (40 ug/mL, 1 h). acetylation level of NLRP3 were analyzed. LE: long exposure; SE: Short exposure

2. There is no mention of AIM2, NLRC4, flagellin, or poly dA-dT in this manuscript. Furthermore, it is necessary to include information about nigericin in the "Introduction" section of the paper.

Answer: According to the suggestions, we added the description of nigericin, a pore-forming toxin derived from *Streptomyces hygroscopicus* to the second paragraph in the "Introduction" section. We also illustrated that Flagellin and poly(dA:dT) are specific agonist for NLRC4 and AIM2 inflammasome activation, respectively, in the "results" section.

3. The presence of the cleaved form of active caspase-1 is a crucial indicator of

inflammasome activation. However, the results presented in the manuscript only show the presence of the proform of caspase-1 and not the cleaved form. This discrepancy needs to be clarified and addressed in order to provide a more accurate assessment of inflammasome activation.

Answer: We have already provided cleaved perform of caspase-1(p10) in the initial manuscript and annotated in the modified version of manuscript.

4. Histological staining for speck protein components, active caspase-1, or mature IL-1 β is necessary to assess inflammasome activation in situ on the tissue shown in Figures 5 and 6. This staining technique would allow for the visualization and localization of these specific components within the tissue, providing valuable insights into the process of inflammasome activation.

Answer: Given to the anti-caspase-1 and anti-IL-1 β antibody can detect both the mature form and the precursors of caspase-1 and IL-1 β , respectively, the histological staining of caspase-1 and IL-1 β in tissue could not distinguish activated inflammasome or not. To solve this question, we extracted proteins from lung tissues in mice challenged with LPS or not, then performed western blot. By this way, precursor and cleaved form(mature) of IL-1 β and caspase-1 can be clearly detected. We added this results in the modified version of manuscript (**Fig2h, 6e, 7e in the revised version**)

5. The NLRP3 oligomerization assay presented in the manuscript is not clearly demonstrated. As seen in other studies, it is necessary to show a distinct ladder-like pattern indicative of NLRP3 oligomerization.

Answer: According to the recent structural findings that NLRP3 is a 12- to 16-mer double-ring cage at resting stage (**Cell**,2021;184(26):6299-6312; **Nature**, 2022;604(7904):184-189), so it can hardly detect a distinct ladder-like pattern of NLRP3 oligomerization like ASC. The oligomerization assay we used in this manuscript is SDD-AGE, which had been already applied in detection of NLRP3

oligomerization (**J Exp Med.**2017,214(11):3219-3238; **EMBO J.** 2023:e113481), the performance of NLRP3 oligomerization all shows smear pattern, but not ladder-like pattern. Furthermore, we performed BN-PAGE, which has also been applied in detection of the oligomerization of NLRP3(**Immunity.** 2023;56(5):926-943; **Nature.** 2016;530(7590):354-7). Consistent with these studies, we observed smear pattern of NLRP3 oligomers in high molecular weight on BN-PAGE and 2D-SDS-PAGE between (**Fig3h-i, 5g-h, S9f-g in the revised version**). Thus, the two NLRP3 oligomerization assays we used both observed smear pattern of NLRP3 oligomerization, consistent with previous studies.

Minor Comments

1. It appears that caspase-1-myc was transfected; however, it needs to be clarified why caspase-1 is not being expressed in the first lane of Figure 1C.

Answer: We apologize for the mistake and corrected it in Figure 1C in the modified version of manuscript. (**Fig1c in the revised version**).

2. The image quality of Figure 2e is low. Is there a specific reason for the observed differences in cell size? Could it be attributed to potential transfection-related damage? Is there a specific rationale for utilizing the Cos7 cell line, a monkey kidney cell line, instead of the HEK293 cell line?

Answer: We replaced the Figure 2e with new data which exhibited more cells in a view with high quality in the modified version of manuscript. In this view, the cell size is almost same, so we thought the differences in cell size in previous Figure 2e do not make sense. We chose Cos7 cell line, instead of the HEK293 cell line, because it has a larger cytoplasm and is suitable for imaging (**Fig3e-f in the revised version**).

3. The target protein of the various inhibitors used in Figure S3a should be mentioned. Additionally, it is essential to assess whether the inhibitors effectively performed their intended functions.

Answer: According to the suggestions, we described the inhibitors in the modified version of manuscript (**Supplementary Fig S4a-Bottom**). However, we thought it is not necessary to assess the intended functions of the inhibitors since we have already demonstrated NU9056 inhibits NLRP3 inflammasome activation and KAT5 is the writer for NLRP3 acetylation. Additionally, the intended function of the inhibitors did not amplify our conclusion.

Reviewer #4 (Remarks to the Author):

Zhang and co-authors claim that acetylation of NLRP3 at lysine 24 by lysine acetyltransferase 5 (KAT5) promotes its oligomerization and inflammasome assembly. This claim is supported by data from NLRP3-reconstituted 293T cells and immortalized bone marrow-derived macrophages as well as by mouse models of LPS-induced endotoxin shock and peritonitis driven by the NLRP3 inflammasome activator, monosodium urate. Multiple approaches, including unbiased proteomics-based mass spectrometry, genetic and pharmacological manipulations are used to firm up the conclusions. Additional strengths of this manuscript include its concise and clear narrative (though attention to some wordings is needed) and the novelty of the finding as regulation of the NLRP3 inflammasome by acetylation has not been extensively studied. However, there are concerns that need to be addressed.

Answer: In accordance with the valuable suggestions, we carefully revised the manuscript and performed several additional experiments. By addressing these comments and concerns, we have strengthened the conclusion of our paper and the physiological relevance of our findings. The point-by-point responses to comments were listed as below.

Major concerns:

- Are NLRC4 and AIM2 acetylated in responses to their respective activators?

Answer: NLRC4 acetylation has been reported earlier (**Theranostics**. 2021 Feb 15;11(8):3981-3995). Thus, we only detected the acetylation of AIM2. We transfected LPS-primed macrophages with poly(dA;dT) to activate AIM2 inflammasome, then performed IP, however, we did not observe the acetylation band in AIM2. (**Figure R3**)

Figure R2 Analysis of AIM2 acetylation

Peritoneal macrophages were treated with or without LPS (100 ng/mL, 3 h) then followed poly (dA:dT) transfection (1 μg/mL, 16 h). Acetylation level of AIM2 were analyzed.

- Are blots such as those shown in Fig.2 A-C been cut and pasted or not? Lanes for the different proteins do not align well.

Answer: The blots showed in Fig.2 A-C are actually intact. To cancel the concerns, we provided the raw data of the western blot in Fig.2 A-C. **(Raw data of Fig3a-c)**

- Indicate where NLRP3 monomer, dimer, etc..., migrate in Fig. 2D. How do the authors explain the diffuse but not distinctive NLRP3 species (monomers, dimers, etc...)? This concern, which also applies to Fig. 4f and Supplementary Fig. 7d, needs to be convincingly addressed as it cast doubt on the scientific rigor.

Answer: According to the recent structural findings that NLRP3 is a 12- to 16-mer double-ring cage at resting stage (*Cell*,2021;184(26):6299-6312; *Nature*, 2022;604(7904):184-189), so it can hardly detected dimers, trimer, or etc., like ASC. The oligomerization assay we used in this manuscript is SDD-AGE, which had been already applicated in detection of NLRP3 oligomerization (*J Exp Med*.2017,214(11):3219-3238; *EMBO J*.2023:e113481), the performance of NLRP3 oligomerization all shows smear pattern, but not ladder-like pattern. Furthermore, we

performed BN-PAGE, which has also been applied in detection of the oligomerization of NLRP3 (*Immunity*. 2023;56(5):926-943; *Nature*. 2016;530(7590):354-7). We observed smear pattern of NLRP3 in high molecular weight on BN-PAGE (**Fig3h-i, 5g-h, S9f-g in the revised version**). Thus, the two NLRP3 oligomerization assays we used both observed smear pattern of NLRP3 oligomerization, consistent with previous studies.

- Quantitative data are needed for Supplementary Fig. 2a, b.

Answer: We provided the quantitative data in the modified version of manuscript (**Supplementary Fig S3c-f in the revised version**)

- The quality of the images in Fig. 5C and Fig. 6C is not optimal.

Answer: We provided the high quality of the images in Fig. 5C and Fig. 6C in the modified version of manuscript (**Fig 6c and 7c in the revised version**)

- Previous studies on NLRP3 acetylation, for example by KAT2B (PMID: 36250925), are not mentioned.

Answer: We added the discussion of KAT2B on NLRP3 acetylation in the modified version of manuscript.

Minor concerns:

- Several phrases (e.g., 2nd sentence of the results) are too long.

Answer: Thanks for the valuable suggestion. We revised several long phrases in the modified version of manuscript.

- Explain what Ack means on the blots of Fig. 1.

Answer: We thank the reviewer for pointing it out. We replaced AcK as acetyl-K to indicate lysine acetylation clearer in the modified version of the manuscript.

- Pay attention to wordings such as inapparent, enzymatical, etc...

Answer: We corrected these wordings in the modified version of the manuscript.

- IL-1 maturation reflects the activity of the NLRP3 inflammasome, but not NLRP3.

Answer: We apologize for the mistake and corrected NLRP3 to NLRP3 inflammasome in the modified version of the manuscript.

REVIEWERS' COMMENTS

Reviewer #1 (Remarks to the Author):

Previous comments have been addressed

Reviewer #2 (Remarks to the Author):

The authors have provided an impressive amount of new in vitro and in vivo data that clarified the questions and critiques I had on the previous version of the manuscript. I would support publication of the manuscript in the current format.

Reviewer #3 (Remarks to the Author):

The authors appropriately addressed the points of concern I had and conducted the necessary experiments.

Reviewer #4 (Remarks to the Author):

The authors have satisfactorily addressed my concerns.

Reviewer #1 (Remarks to the Author):

Previous comments have been addressed

Answer: Thanks

Reviewer #2 (Remarks to the Author):

The authors have provided an impressive amount of new in vitro and in vivo data that clarified the questions and critiques I had on the previous version of the manuscript. I would support publication of the manuscript in the current format.

Answer: Thanks

Reviewer #3 (Remarks to the Author):

The authors appropriately addressed the points of concern I had and conducted the necessary experiments.

Answer: Thanks

Reviewer #4 (Remarks to the Author):

The authors have satisfactorily addressed my concerns.

Answer: Thanks